**Insights on the spatial distribution of global, national and subnational greenhouse gas**
**emissions in the Emissions Database for Global Atmospheric Research (EDGAR v8.0)**
**Authors:** Monica Crippa[2], Diego Guizzardi[1], Federico Pagani[2], Marcello Schiavina[6], Michele
Melchiorri[1], Enrico Pisoni[1], Francesco Graziosi[1], Marilena Muntean[1], Joachim Maes[5], Lewis
Dijkstra[1,5], Martin Van Damme[3,4], Lieven Clarisse[3], Pierre Coheur[3]
[1]European Commission, Joint Research Centre (JRC), Ispra, Italy
[2]Unisystems S.A., Milan, Italy
[3]Spectroscopy, Quantum Chemistry and Atmospheric Remote Sensing (SQUARES),
Université libre de Bruxelles (ULB), Brussels, Belgium
[4]Royal Belgian Institute for Space Aeronomy (BIRA-IASB), Brussels, Belgium
[5]European Commission, Directorate-General for Regional and Urban Policy, Brussels,
Belgium
[6]NTT DATA, Rue de Spa, 8, 1000 Brussels, Belgium
Correspondence: enrico.pisoni@ec.europa.eu
**Abstract**
To mitigate the impact of greenhouse gas (GHG) and air pollutant emissions, it is of the utmost
importance to understand where emissions occur. In the real world, atmospheric pollutants are
produced by various human activities from point sources (e.g. power plants and industrial
facilities) and also from diffuse and area sources (e.g. residential activities and agriculture).
However, as tracking all these single sources of emissions is practically impossible, emission
inventories are typically compiled using national-level statistics by sector, which are then
downscaled at the grid-cell level using spatial information. In this work, we develop high-
spatial-resolution proxies for use in downscaling the national emission totals for all world
countries provided by the Emissions Database for Global Atmospheric Research (EDGAR).
In particular, in this paper we present the latest EDGAR v8.0 GHG, which provides readily
available emission data at different levels of spatial granularity, obtained from a consistently
developed GHG emission database. This has been achieved through the improvement and
development of high-resolution spatial proxies that allow the more precise allocation of
emissions over the globe.
A key novelty of this work is the potential to analyse subnational GHG emissions over the
European territory and also over the United States, China, India and other high-emitting
countries. These data not only meet the needs of atmospheric modellers but can also inform
policymakers working in the field of climate change mitigation. For example, the EDGAR
GHG emissions at the NUTS 2 level (nomenclature of territorial units for statistics level 2)
over Europe contribute to the development of EU cohesion policies, identifying the progress
of each region towards achieving the carbon neutrality target, as well as providing insights on
the most highly emitting sectors. The data can be accessed at
https://doi.org/10.2905/b54d8149-2864-4fb9-96b9-5fd3a020c224 specifically for EDGAR
v8.0 (Crippa, 2023a) and https://doi.org/10.2905/D67EEDA8-C03E-4421-95D0-
0ADC460B9658 for the subnational dataset (Crippa et al., 2023b).

## 1. Introduction

Knowing where emissions are released is essential to support the design of effective mitigation actions and for atmospheric modelling purposes. Emission inventories are typically developed at the national level and provide sector-specific emission estimates. In order to disaggregate national emissions over high-resolution grids, information on the location of the different emission sources (e.g. point, linear and area sources) must be collected, and 'spatial proxies' should be developed and applied to national sector-specific emission totals to downscale them over grid maps. The correct allocation of point source emissions is essential to avoid misplacing high emission levels. However, gathering information on point sources covering the entire globe and a wide temporal domain (1970 to present) is challenging because of limitations in data availability, in the accuracy of the reporting (real location vs legal address, etc.) and in the completeness of data.

The Emissions Database for Global Atmospheric Research (EDGAR) provides global greenhouse gas (GHG) and air pollutant emissions over the global grid map at $0.1° \times 0.1°$ resolution, obtained through a downscaling process of national emissions using high-resolution spatial data. The development and maintenance of the EDGAR grid maps is essential, since several regional and global databases rely on the EDGAR emission grid maps to disaggregate national emissions to the grid. This is the case for the Community emissions data system (Feng et al., 2020; Hoesly et al., 2018) or the European monitoring and evaluation programme (EMEP) Centre on Emission Inventories and Projections, which supports Parties to the Convention on Long-range Transboundary Air Pollution in meeting their official gridded emission reporting obligations (CEIP, 2021).

This work is an update of previous EDGAR publications dealing with spatial data (Janssens-Maenhout et al., 2019; Crippa et al., 2021), and describes all the new developments in the spatialisation of the emissions from EDGAR v8.0 onwards, focusing on high-emitting sectoral point sources, such as power plants and industrial activities, but also on more diffuse sources such as residential activities. High-resolution spatial information has been gathered at the global level by combining data from the Global Energy Monitor, official registries and satellite retrievals. The relevance of using updated spatial information is also assessed through regional case studies.

The purpose of this publication is to describe the EDGAR v8.0 GHG gridded emission datasets, focusing on the updates to the spatial proxies included in this data release. The analysis of EDGAR v8.0 emission time series (European Union, 2023; IEA-EDGAR $CO_2$, 2023) and the methodology behind emission calculations is available in Crippa et al. (2023c).

The main novelties of this work are (i) an update on emission point sources using global datasets (e.g. Global Energy Monitor), (ii) the development of a gap-filling method for non-population-based sources using built-up surface information for non-residential areas(*) from the Global Human Settlements Layer (GHSL), (iii) an update of population-based proxies using

---

(*)    This information is compliant with the definition of 'building' as per the infrastructure for spatial information in Europe (Inspire) directive (https://inspire.ec.europa.eu/id/document/tg/bu) for non-residential areas (e.g. industrial or commercial facilities, warehouses) from the Global Human Settlements Layer.

the latest GHSL data, including a weighting for the temperature-dependent need for heating,
and (iv) an update on international ship tracks and weights by vessel type. In addition,
information at the subnational level (e.g. for Europe at the NUTS 2 level (nomenclature of
territorial units for statistics level 2)) is included when developing the new spatial proxies for
EDGAR, thus allowing a more accurate allocation and analysis of subnational emissions. The
EDGAR v8.0 GHG global emission maps can be accessed at
https://doi.org/10.2905/D67EEDA8-C03E-4421-95D0-0ADC460B9658 for the subnational
emissions and at https://doi.org/10.2905/b54d8149-2864-4fb9-96b9-5fd3a020c224 for v8.0
for the emission grid maps at $0.1 \times 0.1°$ resolution.
**2. Overview of the methodology and data sources used for updating spatial information**
**in EDGAR**
Bottom-up global inventories (such as EDGAR) compute emissions for each sector, pollutant
and year at the national level, making use of international statistics and official guidelines for
emission computation (Janssens-Maenhout et al., 2019; Crippa et al., 2018). However,
atmospheric modellers, policymakers, local authorities and scientists may need to analyse
spatially distributed emissions at a higher resolution than country-level data. Therefore, annual
country-specific emissions are distributed over the globe making use of spatial information,
representing the exact location of point sources (e.g. power plants, industrial facilities), linear
tracks (e.g. road network, ship and aeroplane tracks) or area sources (e.g. populated areas,
industrial areas). Within the EDGAR database, over 130 proxy datasets ($f$) varying over time
are developed to distribute the contribution of sector-specific emissions ($EM_{i,j,k}$) of each
country ($C$) and pollutant ($x$) over time ($t$) to each grid cell ($em_{i,j,k}$) at $0.1° \times 0.1°$ resolution
(about 10 km spatial resolution at the equation, considering the World Geodetic System
WGS84, EPSG:4326). The Heaviside function (i.e. unit step function whose value is zero for
negative arguments and 1 for positive arguments) is also used, equalling 1 when the grid cell
belongs to the country area, accordingly with the following formula:
$$em_{i,j,k}(lon, lat, t, x) = EM_{i,j,k}(C, t, x) \cdot \frac{f_{i,j,k}(lon, lat, t)}{\sum_{lon,lat}(f_{i,j,k}(lon, lat, t) \cdot H_{i,j}(C, lon, lat))},$$

where
$H_{i,j}(C, lon, lat)$ = fraction/weight of grid cell within $C$,
$i$ = sector,
$j$ = fuel,
$k$ = technology.
Table 1 summarises the data sources and the methodology used to update spatial information
for each emitting sector in the EDGAR database, highlighting the most relevant and latest
updates compared with previous EDGAR data releases. These updates apply from EDGAR
v8.0 onwards. Being a global database of emissions, the spatial data sources are typically
developed at the global level (e.g. satellite-based retrievals) but often rely on national data
collection (e.g. national point source information reported to fulfil legal requirements).
Therefore, the same data sources may be used by other inventory developers to update their
spatial disaggregation of the emission data. In the following sections, a detailed description of
the data sources and the approach used for updating each emission sector is provided,
distinguishing between point sources, area sources and linear sources. For all sectors not
subjected to a recent revision in the EDGAR database, we refer the reader to the overview
Table S1 in the Supplement and the references therein.
A key methodological advance in the EDGAR gridding system is the inclusion of subnational
attributes for each spatial proxy and in particular for each point source. This implies attaching
to each point not only its exact location, expressed in longitude and latitude, but also the related
NUTS 2 code (EUROSTAT, 2021) for Europe or the Global ADMinistrative layer at level 1
(GADM version 4.1). The decision to include NUTS 2 rather than NUTS 3 information aims
to enhance the capability of a global database such as EDGAR to represent subnational regional
emissions in support of the development of regional policies (e.g. EU cohesion reports
(European Commission, 2022)) or the 2040 climate impact assessment. The attribution of
subnational details is developed not only with an EU-oriented focus but also for other countries
such as China, India and the United States by providing emissions at the state or province level.
The purpose of our work is to provide readily available emission data at the subnational level
estimated in a consistent way for all countries. The EDGAR data may represent an
approximation for those countries with a developed statistical infrastructure (e.g. those
including subnational statistics and very precise spatial proxies); however, they provide a
default if such data are not available, as is the case for many countries in the world. In the
results section, case studies on subnational emissions are presented for the EU, China, India
and the United States.
**3. Point sources of emissions**
Gathering information on point sources covering the globe and spanning a wide temporal
domain (1970 to present) is challenging because of the limited data available and their accuracy
and completeness in the reporting (real plant location vs legal address, etc.). Establishing the
correct location of point sources is essential, since they are often super-emitters (e.g. power
plants for $CO_2$ emissions). In EDGAR v8.0, the locations of the main industrial point sources
(e.g. power plants, iron and steel industries, coal mines, venting and flaring activities), which
contribute around half of global $CO_2$ emissions, have been updated using state-of-the-art
information from global databases, such as the Global Oil and Gas Plant Tracker and Global
Coal Plant Tracker of the Global Energy Monitor. A complete overview of the data sources
and updates included in EDGAR v8.0 is provided in Table 1.
However, point source databases are characterised by some limitations, such as the
completeness of information on the point sources, the availability of time series for
information, the misplacement of data points compared with their actual country location, etc.
In EDGAR v8.0, quality control procedures are applied to validate the correct location of each
point source to the corresponding country or subnational attribute. Moreover, missing
information is completed using assumptions on the lifetime of power plants (i.e. 40 years) to
indicatively attribute the opening or closing years for each plant.
No consistency checks between $CO_2$ emissions estimated using independent methods have
been performed. here However, Guevara et al. (2024) have proven that there is good agreement
between national $CO_2$ emissions from power plants reported by EDGAR (which are based on
international statistics) and plant-level inventories.
Atmospheric modellers require information not only on the spatial patterns of the emissions
but also on their temporal and vertical distribution, as described in Ahsan et al. (2023), Bieser
et al. (2011) and de Meij et al. (2006). For example, de Meij et al. (2006) found that the vertical
distribution of emissions of $SO_2$ and nitrogen oxides ($NO_x$) plays an important role in
understanding the differences between emission inventories in calculated gas and aerosol
concentrations. Accordingly, in the EMEP model, industrial point source and power plant
emissions occur in up to the third level (top up to 184 m), while shipping emissions happen in
the first level (top up to 20 m). However, addressing the vertical distribution of the emissions
in beyond the scope of this work. In the following sections, we will describe sector by sector
how the most up-to-date spatial data on point sources have been collected and implemented in
the EDGAR database to downscale national emissions over the global grid map.

## 3.1. Power plants

Power plants represent a major source of fossil fuel-derived $CO_2$ and other GHG emissions
globally, nowadays contributing around 38 % and 18 %, respectively, of the corresponding
global totals (Crippa et al., 2023c). It is therefore of utmost importance to spatially allocate
these emissions correctly at the global level and understand their trends over time, in order to
design and implement adequate emission mitigation measures.
In EDGAR v8.0, fuel-specific spatial proxies have been developed using data from the Global
Coal Plant Tracker and Global Oil and Gas Plant Tracker of the Global Energy Monitor (for
coal and gas) (Global Energy Monitor, 2022b, c), the Global Power Plant Database v1.3.0
(World Resources Institute, 2018; WRI, 2021) for oil and biofuels, the Carbon Monitoring for
Action database (CARMA v3.0) for autoproducers (i.e. plants and industries producing power
for their own use). In addition, information on autoproducers and biofuel-fired power plants in
Europe has been integrated using the European Pollutant Release and Transfer Register
(EPRTR v18) (EPRTR, 2020). For the US domain, the location of fossil fuel-fired power plants
is taken from the US Energy Information Administration (US EIA, 2022b), as it represents the
most up-to-date source for the United States. The time frame covered by the new power plant
spatial proxy datasets developed in EDGAR v8.0 is 1970–2022, which includes, for each plant,
information on opening and closing years (including beyond 2022 for recently built power
plants), capacity, main fuel type, etc. When only partial information is available for the years
of operation, assumptions based on the typical lifetime of power plants are made (e.g. 40 years).
The capacity of each power plant is used to relatively weight within a country the fuel-specific
emissions from power plants. An additional adjustment is performed for the US data to account
for the different sulphur content in the fuel used in different US states based on EIA and Federal
Energy Regulatory Commission utility surveys.
The Global Energy Monitor is chosen as the main data source for updating power plant proxies,
since it relies on data from public and private data sources (including the Global Energy
Observatory, CARMA, Platts World Energy Power Plant database, national-level trackers
developed by environmental organisations, and various company and government sources). It
is validated with (i) government data on individual power plants, (ii) country energy and
resource plans and government websites tracking coal plant permits and applications, (iii)
reports by state-owned and private power companies, (iv) news and media reports, and (v) local
non-governmental organisations tracking coal plants or permits. Local experts are also
involved in the review of coal and gas plant data. Regular biannual updates of these databases
also guarantee the possibility of including further updates in future EDGAR releases. As of
January 2019, the Global Coal Plant Tracker included the exact locations of 95.3 % of
operating units (6 411 out of 6 725). Independent use and validation of the Global Coal Plant
Tracker and Global Oil and Gas Plant Tracker is also performed by Guevara et al. (2024).
Figure S1 in the Supplement shows the comparison between the geographical coverage of
EDGAR v8.0 and the previous EDGAR spatial data for power plants, while Figure S2 provides
a view of the global coverage of power plants in EDGAR v8.0 by fuel type.
Figure 1 shows the global coverage and intensity of $CO_2$ emissions from fossil fuel-fired power
plants from EDGAR v8.0 for the years 1970 and 2022. As a general trend, the number of power
plants increased strongly from 1970 to 2022 (see also Figure 2) due to global industrialisation
over those five decades, although the number of power plants in 1970 is more uncertain than
that for the present day.
The total number of power plants grew from around 8 500 in 1970 to 13 000 in 2022, with the
sharpest increases occurring in China (4.5 times more) and North America (2 times more).
However, the intensity of the emissions has changed over the past five decades, depending on
the region. As shown in Figure 2, despite the increase in the regional number of power plants,
the shift towards cleaner fuels in historically industrialised regions (such as Europe and North
America), together with increased energy efficiency, has led to stable and lower $CO_2$ emissions
in these regions (e.g. a 13 % decrease in emissions in Europe between 1970 and 2022). In
contrast, emerging regions are characterised by significantly higher emissions in 2022 and the
use of high-carbon-content fuels, such as coal. Over the past five decades, fossil $CO_2$ emissions
from power plants have increased up to 42 and 38 times in China and India, respectively.
Country-specific trends in $CO_2$ and GHG emissions from power plants are presented in Crippa
et al. (2023c).

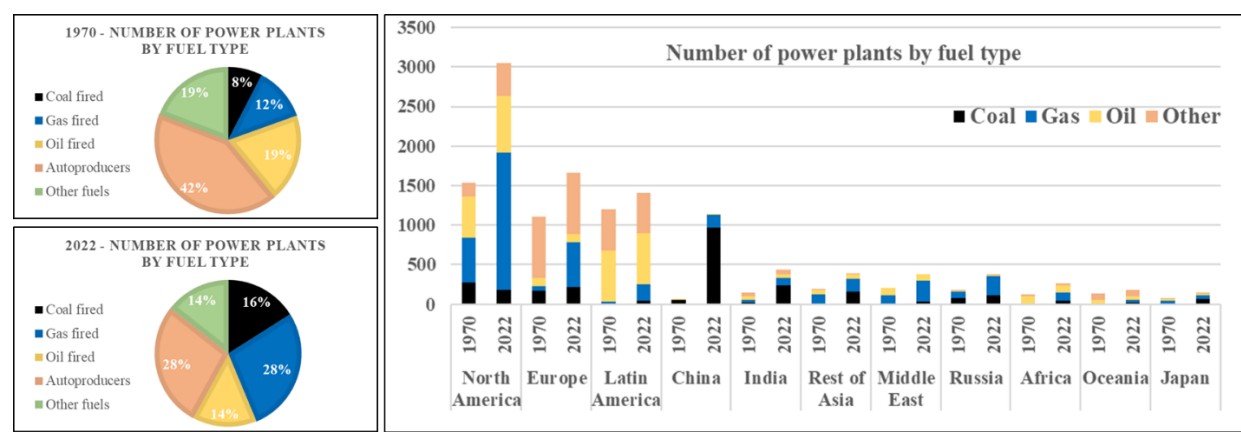


**Figure 1 – CO₂ emissions from fossil fuel-fired power plants in 1970 and 2022 from EDGAR v8.0. The size of the circles is proportional to the magnitude of the emissions.**


**Figure 2 – Increase in the total number of power plants (including fossil fuel- and biofuel-fired plants) from 1970 to 2022 by world region, as included in the updated EDGAR spatial proxies.**



## 3.2. Industrial facilities and other point sources

Industrial activities cover a wide range of sectors encompassing the production of iron and steel, cement, glass, metals, chemicals and fertilisers and the use of solvents but also intensive animal farming (see Section 3.4). Gathering information on industrial activities (e.g. production, capacity, location of the facilities) at the global level is challenging, in part because of confidentiality and data protection issues. For this reason, we focused not only on the updating of information on industrial point sources (when available) but also on improving the gap-filling method for all industrial activities if data are incomplete or missing (as discussed in detail in Section 3.5). In EDGAR v8.0, we included the latest EPRTR (EPRTR v18) locations for all industrial facilities (with the exception of power plants, iron and steel facilities, and coal mines, for which dedicated spatial proxies have been developed at the global level). Several manual adjustments were made to overcome data quality issues related to missing spatial information and inconsistencies. The analysis of the EPRTR dataset also inspired the idea of attributing only a fraction of the emissions to the reported point sources. This is justified by the fact that industrial facilities have to report their emissions only if they fall above a certain threshold. The fraction of the emissions to be allocated to the available point sources is determined through the ratio between the EPRTR emissions (typically of $CO_2$) and the corresponding EDGAR emissions. When the ratio is 1, all emissions are allocated to the point sources; when the ratio is lower than 1, the complementary fraction is then attributed to the gap-filling grid (i.e. non-residential proxy as defined in Section 3.5).

In EDGAR v8.0, we have also updated the global locations of iron and steel plants, which are among the most energy-intensive industries. The Global Steel Plant Tracker of the Global Energy Monitor (2022d) was used as a data source because of its global and temporal completeness (1970 to present). The installed capacity was used to weight the relative contribution of each iron and steel plant, although it may represent an approximation of the real capacity in use. A map of iron and steel production plants in 1970 and 2022 is presented in Figure 3. The number of iron and steel plants increased around 10-fold over the last five decades (from 77 to 728) with the sharpest increases in China (5-fold) and the United States and India (2.7-fold).

Coal mines are also a relevant source of fugitive emissions of GHGs and air pollutants (e.g. volatile organic compounds). In EDGAR v8.0, we updated the information on coal mines at the global level using the Global Coal Mine Tracker of the Global Energy Monitor (2022a) complemented with the EIA data for the United States (US EIA, 2022a). For countries not covered by these data sources, we relied on the previous EDGAR spatial proxies including data from the United States Geological Survey (USGS, 2019). More specifically, we included information on surface and underground mines for both hard and brown coal.

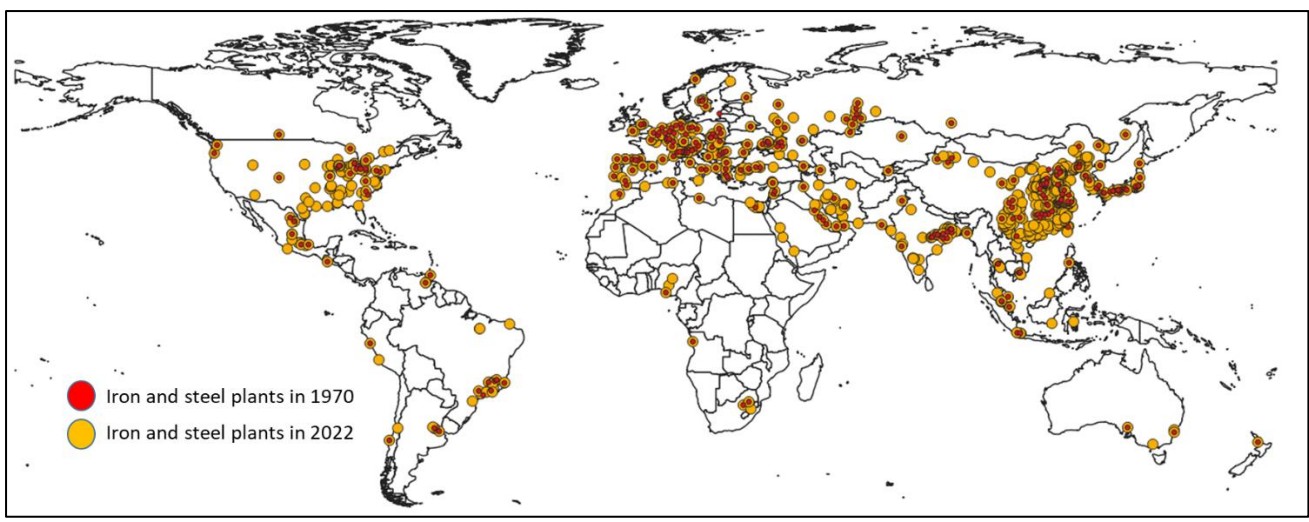


**Figure 3 – Global locations of iron and steel plants in 1970 and 2022.**
**3.3. Venting and flaring**
Gas flaring is the burning of the natural gas that results from oil extraction. Although this
practice is highly polluting and represents a waste of resources, it still takes place in several
countries because of economic constraints and a lack of appropriate legislation. Flaring takes
place at both onshore and offshore installation, and it is a source of GHG and air pollutant
emissions.
Global $CO_2$ emissions related to flaring accounted for 276 Mt in 2022, of which 76 % was
emitted by 10 countries, namely Russia (18 % of the global total), Iraq (13 %), Iran (12 %) and
Venezuela (7 %), followed by Algeria, United States, Mexico, Libya, Nigeria and China.
Although this emission source represents only 0.8 % of global $CO_2$ emissions, it is particularly
relevant for certain regions of the world, such as Venezuela (20 % of the country's total $CO_2$
emissions), Iraq (18 %), Libya (17 %), Algeria (10 %) and Nigeria (9 %). Considering the
relevance of venting emissions and the potential for control measures, it is essential to
accurately quantify and attribute this source to the correct location. Flaring emissions can also
be localised and quantified using spaceborne measurements (Elvidge et al., 2017; NOAA,
2017). In EDGAR v8.0, data from the World Bank *Global Gas Flaring Tracker Report* (2023)
were used for estimating both the emissions and the location of global flaring activities from
2012 to 2022. These spatial data were also used as a best approximation to spatially distribute
emissions from venting, which is the controlled release of natural gas without it being burned,
although the two activities may not overlap. The resulting map of $CO_2$ emissions in 2012 and
2022 is shown in Figure 4.

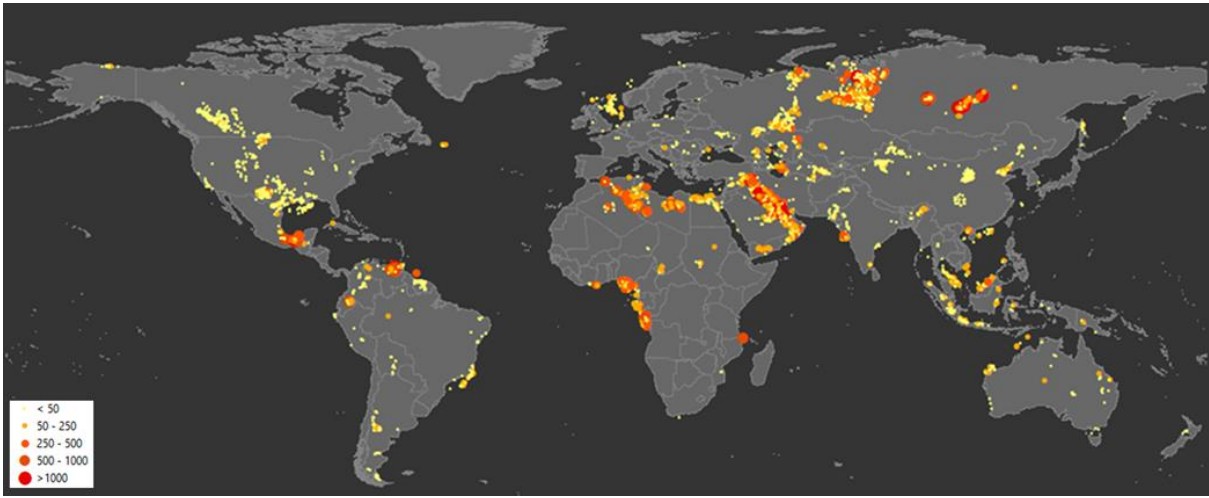

**Figure 4 – Global map of CO₂ emissions (kt) from flaring in 2022.**

### 3.4. Intensive livestock and fertiliser-manufacturing industries

Agriculture includes a variety of activities that are typically distributed over large areas (e.g. crop areas, animal pastures). However, several agricultural activities can be defined as hotspots or point sources and include intensive animal farming and manure management practices. In a broader sense, we also allocate to this sector the fertiliser-manufacturing industry, which represents an important source of $NH_3$ and $N_2O$. In EDGAR v8.0, the infrared atmospheric sounding interferometer (IASI) satellite-derived $NH_3$ point source database (Van Damme et al., 2018; Clarisse et al., 2019) is included to map emissions from animal farming and fertiliser production with yearly information for the period 2008–2022. It includes 270 agricultural hotspots and 251 synthetic $NH_3$ production facilities worldwide. Since the $NH_3$ point source database includes only hotspots, we decided to allocate to these points only a fraction of the total emissions for that sector and country derived from approximate estimates of $NH_3$ emission fluxes from IASI measurements, while distributing the remaining fraction to livestock density maps formerly available in EDGAR. Similarly to what was done for other industries, for Europe, intensive livestock and fertiliser production point sources were taken from EPRTR v18. Similarly, the satellite-based information on fertiliser industries was integrated into the previous EDGAR proxy for this sector. This update represents a significant improvement in representing nitrogen-related hotspots (Van Damme et al., 2018) compared with earlier EDGAR releases which mostly used animal density as a proxy (see Table S1), albeit taking into account that the uncertainty of IASI information is around 50 %. A snapshot of $N_2O$ emissions from manure management at the global level and in Europe, where intensive livestock activities appear as emission hotspots, is shown in Figure 5.

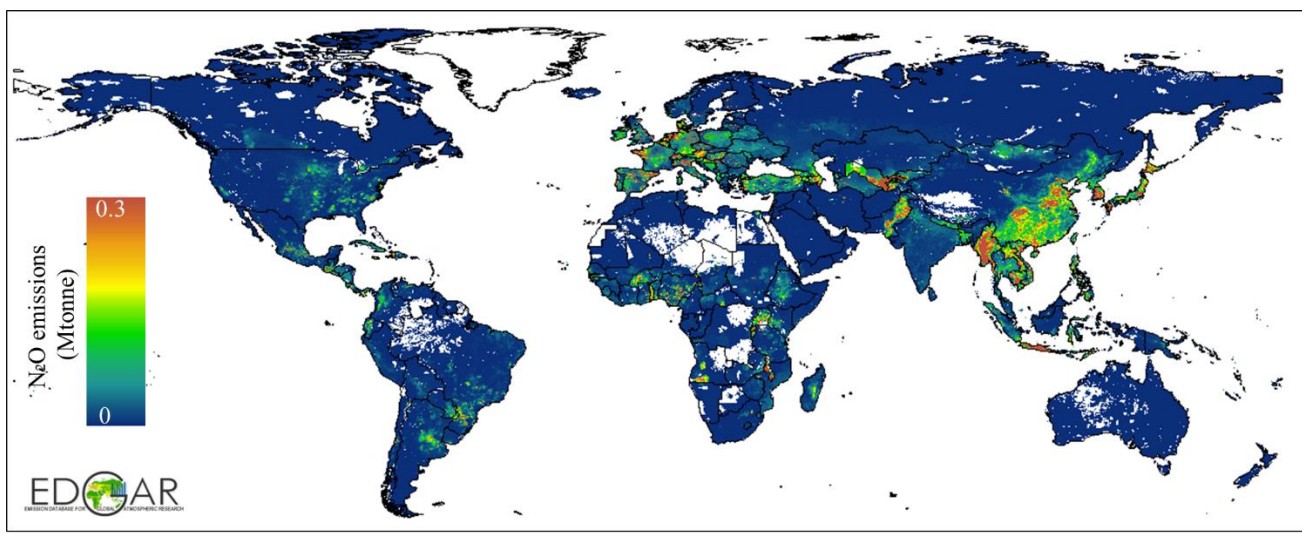

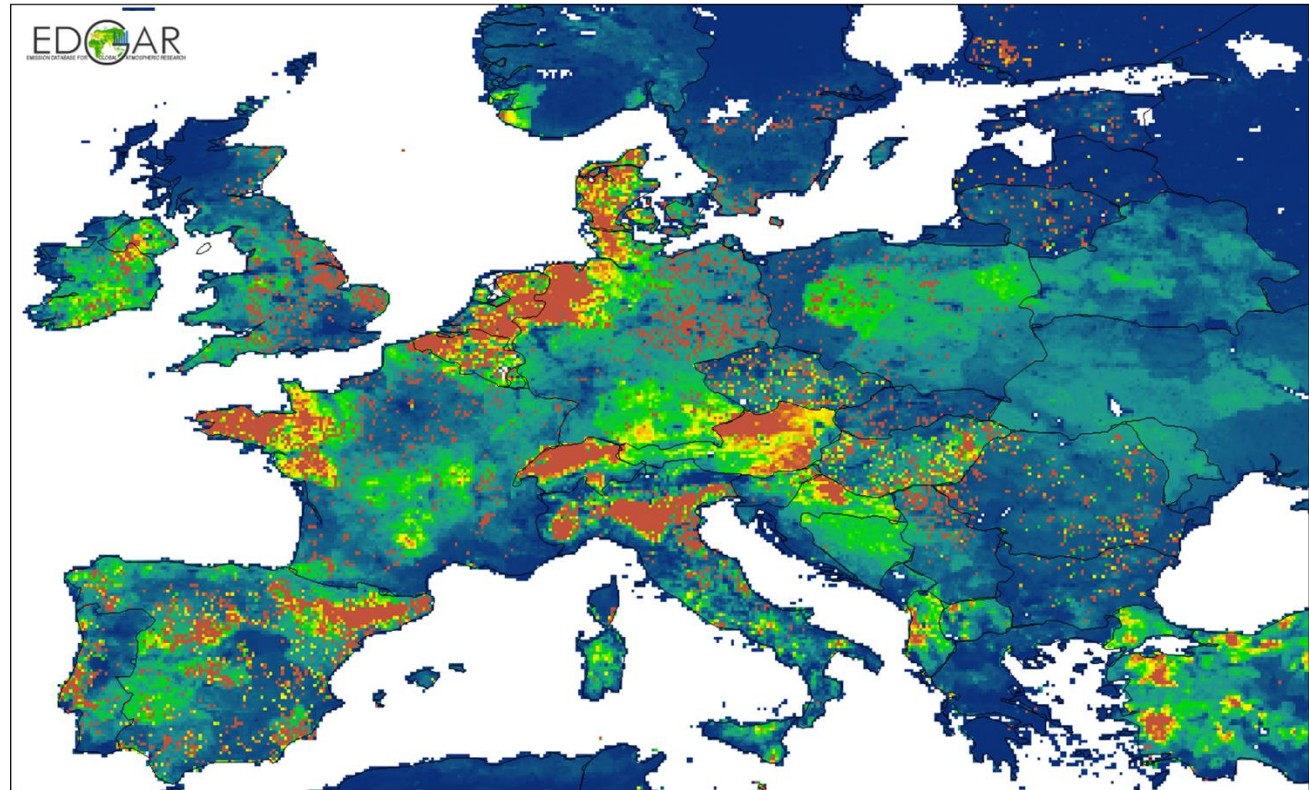


**Figure 5 – N₂O emissions from manure management at the global level and in Europe, where intensive**
**livestock activities appear as emission hotspots.**

### 3.5. Gap filling missing information for point sources

A significant improvement is represented by the development and use of a new spatial proxy
to gap fill missing information for all industry-related emissions. Until EDGAR v7.0,
population-related proxies were used as backup information when no spatial data were
available to represent the emissions for a sector within a country (Crippa et al., 2021). However,
here we decided to use the non-residential built-up surface information developed by the GHSL
(Pesaresi and Politis, 2023; European Commission, 2023) as a backup proxy to distribute the
emissions of all the activities not related to small-scale combustion for which no point source
information was available (even for individual countries). This methodological assumption is
a key novelty of this work because of its application at the global level. However, it is in line
with methodologies already applied in regional inventories, such as in Europe (Kuenen et al.,
2022), where the Corine Land Cover dataset is used to spatially allocate emissions to areas
with industrial activity, thus supporting the validity of this assumption.
For certain sectors and regions, this non-residential gap-filling proxy is also used to allocate a
fraction of the emissions of certain sectors (see, for example, the industrial facilities section for
Europe). The overall effect of using this new proxy is a change in the industrial contribution
over densely populated areas, which was previously higher in EDGAR than in other inventories
over Europe in particular (Thunis et al., 2023). Figure 6 shows $CO_2$ emission maps from
manufacturing industries obtained from EDGAR v7.0 and v8.0. This figure highlights the
implications of using different gap-filling proxies for the industrial sector and in particular
contrasts those based on population (EDGAR v7.0) with the new ones based on non-residential
built-up surface data (EDGAR v8.0).
Overall, using non-residential built-up information to allocate emissions of industrial activities
to complement point source information leads to lower emission levels being allocated to urban
areas and a less densely distributed map over certain regions (e.g. China, India). Figure S3
shows the impact of this update on global fossil fuel-derived $CO_2$ emissions from the industrial
sector over global functional urban areas (FUAs) in 2022. The share of $CO_2$ industrial
emissions of the national total over FUAs is typically higher, on average by around 30 %, in
EDGAR v8.0 than in EDGAR v7.0 for several developing countries (e.g. Africa, India, South
America) because of the presence of industrial point sources and non-residential activities still
close to urban areas. However, lower emissions from industries (on average around 20 % less)
are found in many industrialised regions (e.g. Europe, Oceania, United States) because of the
displacement of industrial activities in remote areas or outside the FUAs. This result represents
the effect of using non-population-based proxies for industrial emissions in EDGAR v8.0
compared with previous EDGAR proxies.
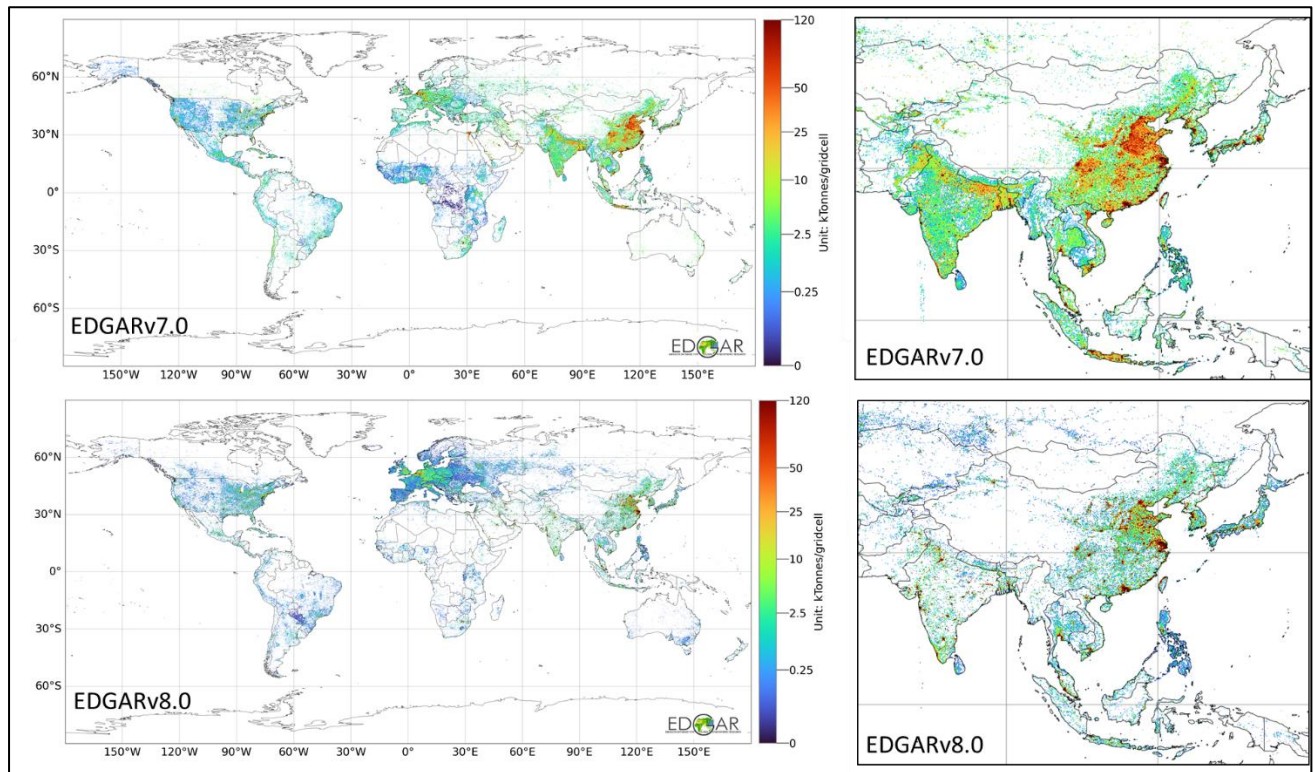

**Figure 6 – CO$_2$ emissions from industrial combustion in 2021 from EDGAR v7.0 (top) and v8.0 (bottom),**
**showing the impact of the gap-filling proxies used for industrial sources.**

## 4. Linear sources of emissions: international shipping

Since EDGAR v6.0, international shipping emissions have been distributed using the Ship
Traffic Emission Assessment Model (STEAM 3) from the Finnish Meteorological Institute
(Jalkanen et al., 2012; Johansson et al., 2017) and this approach has remained unchanged in
EDGAR v8.0. Emissions are distributed on a yearly basis from 2000 to 2018, including multi-
vessel information (cargo, container, fishing, passenger cruiser, service, tanker, vehicle carrier,
miscellaneous). Compared with the previous EDGAR proxy, the use of the STEAM data allows
a better representation of the trend over time in international shipping emissions, differentiating
on an annual basis the variation in the routes and their intensity for the different vessels
consistently with the information available in EDGAR (see Figure 7). Only data covering sea
areas are included, since inland data over big rivers or lakes is not robust enough to be included
in EDGAR. Information on emission control areas, and in particular on sulphur emission
control areas and NO$_x$ emission control areas, is not yet included, although this may be
considered in future updates of EDGAR. A comparison of the international shipping intensities
available in EDGAR before and after this update is presented in Figure S4 of the Supplement.
Figure 8 focuses on three main vessel types representing the largest fraction of GHG emissions
from international shipping in 2022 and contributing specifically around 22 % (tankers), 24 %
(containers) and 28 % (cargo) of total international shipping GHG emissions. The impact of
using the STEAM data to develop the new spatial proxies for international shipping is shown
in Figure 8, which presents a comparison between EDGAR v5.0 and EDGAR v8.0 CO$_2$
emissions from the three main vessel types over the different oceans and seas. EDGAR v5.0
used an in-house EDGAR proxy based on Wang et al. (2008), improved with long-range
identification and tracking information (Alessandrini et al., 2017) for European seas, as
described in Janssens-Maenhout et al. (2019). EDGAR v5.0 proxies were allocating most of
the international shipping emissions over the Atlantic and Pacific Oceans, while the new
proxies of EDGAR v8.0 allocate the largest portion of these emissions (40 %) over the seas
around China, Japan and the Philippines. The relative share of tanker emissions over the
Mediterranean Sea is also very different between the two versions, with the largest contribution
(85 %) from the three categories considered in EDGAR v5.0. Emissions allocated to the Gulf
of Mexico and Arabian Sea are two times higher using the STEAM-based proxies in EDGAR
v8.0.


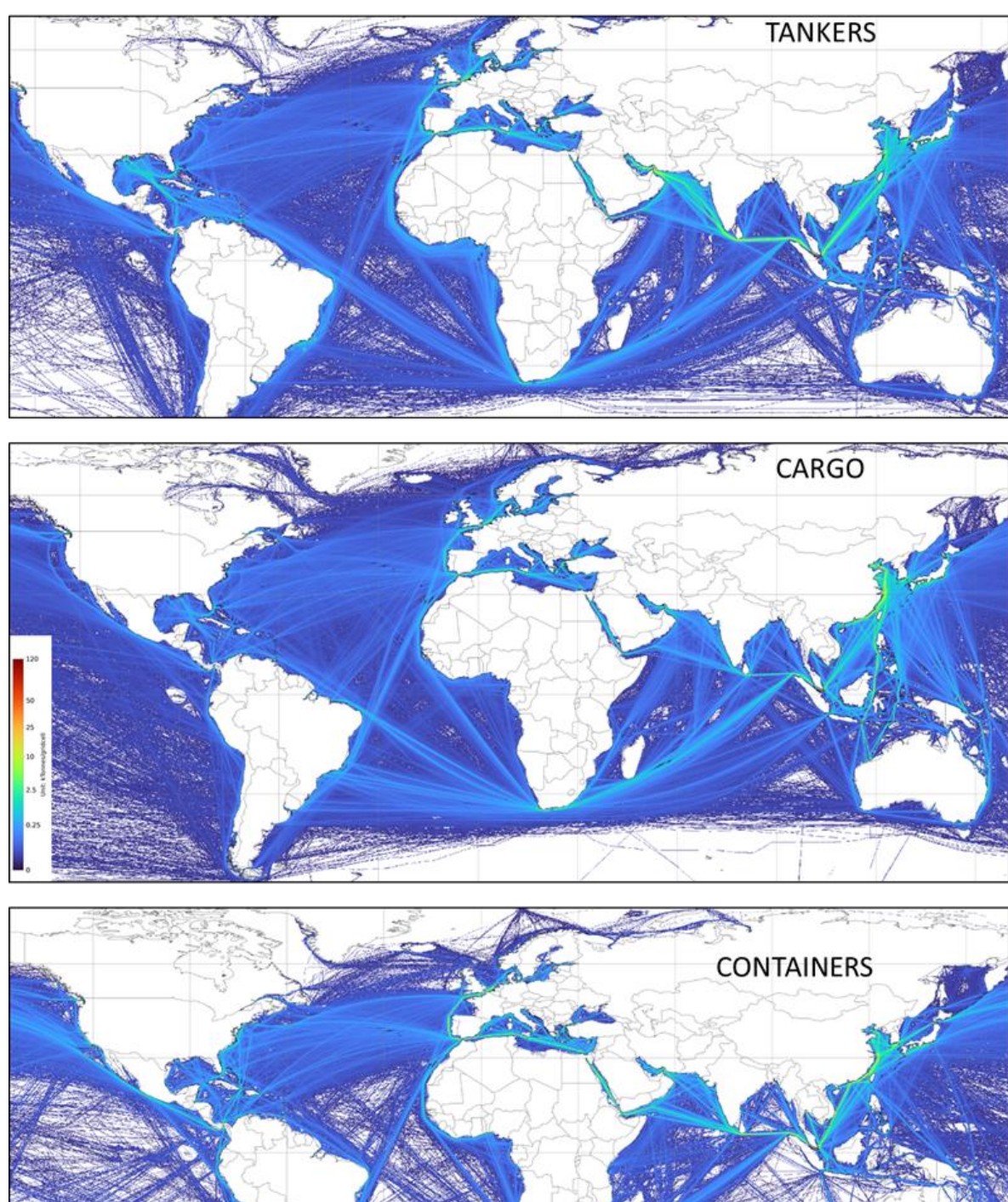

**Figure 7 – International shipping GHG emissions in 2021 showing the ship tracks for tankers, cargo vessels and containers as in EDGAR v8.0.**

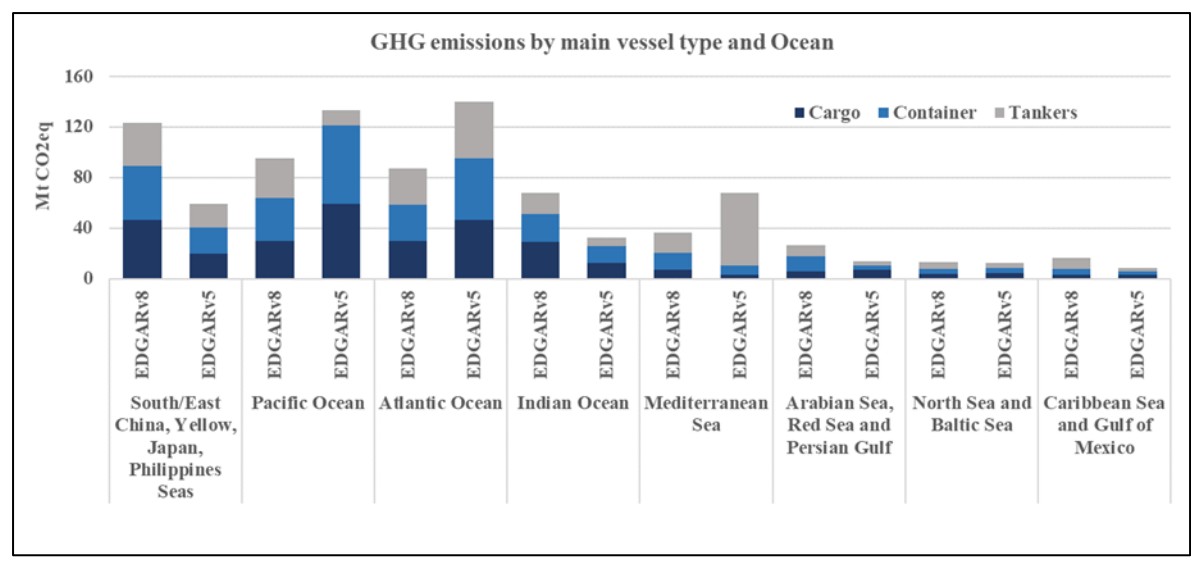


**Figure 8 – Comparison of GHG emissions from international shipping in 2022 by main vessel type and**
**ocean or sea from EDGAR v5.0 and v8.0. Fishing-, service- and passenger-related emissions are excluded**
**from this comparison.**
## 5. Area sources of emissions
### 5.1. Residential activities
Small-scale combustion emissions are mostly related to non-industrial activities, such as those
from the residential, commercial, agricultural and fishing sectors. Therefore, population-based
spatial proxies are often used to downscale national emissions. EDGAR v8.0 aims to couple
population distribution with heating degree-days, since the amount of emissions is not only
dependent on the number of people living in a certain area but also on the meteorological
conditions and the need for heating of indoor spaces. Residential emissions are therefore
distributed considering both population intensities and heating needs, with varying profiles
from 1970 to 2022. EDGAR v8.0 includes the latest population grid maps developed by Global
Human Settlements, GHS-POP R2023A (Schiavina et al., 2023b; Freire et al., 2016), which
comprise residential population information for 12 epochs, over 1975–2020 with 5-year time
steps and projections to 2025 and 2030 obtained by distributing census data from CIESIN
GPWv4.11 over global grid maps. GHS-POP R2023A data at 30 arc-seconds (WGS84,
EPSG:4326) (or about 1 km) spatial resolution were used to develop the corresponding spatial
proxies in EDGAR. Population density is then calculated for each grid cell and used as a proxy
to allocate household emissions over populated areas. Small-scale combustion activities related
to agriculture are distributed using rural population maps obtained from the GHS-SMOD
R2023 product (including only low- and very low-density rural grid cells) (Schiavina et al.,
2023a). For missing years, the closest population map to each epoch is taken (e.g. for the years
2001 and 2002 the population map from 2000 is used, while for the years 2003 and 2004 the
2005 map is used).
To account for the effect of the weather (ambient temperature) on heating needs in the
residential sector, heating degree-days (HDDs) were computed using the 2 m surface air
temperature data with hourly time resolution and 1° spatial resolution using the Copernicus
ERA5 atmospheric reanalysis produced by the European Centre for Medium-Range Weather
Forecasts for the years 1970–2022
(https://cds.climate.copernicus.eu/cdsapp#!/dataset/reanalysis-era5-single-levels?tab=form).
HDDs is the cumulative number of degrees by which the mean daily temperature falls below a
reference temperature (usually 18 °C or 19 °C, which is adequate for human comfort). HDDs
were calculated following the methodology described by Spinoni et al. (2018) and assuming a
reference temperature of 18 °C. Cooling degree-days are not included in the development of
the spatial proxies, since they are mainly related to electricity consumption rather than to fuel
combustion in the residential sector. An additional weight is therefore added to the population
distribution by using the HDD metric, thus increasing the emissions arising in colder regions
with a greater need for heating than in warm areas for the same amount of population.
Our approach does not aim to identify and represent heating habits for all countries but within
a single country modulates the differences in combustion of fuels for, for example, heating
purposes due to the different mean temperatures across latitudes (climatic zones). Country
populations may also have different habits in terms of turning on and off their heating systems,
thus requiring the use of different reference temperature values in the calculation of HDDs
(Atalla et al., 2018), which is not taken into account here. The process of building the residential
proxy in EDGAR is shown in Figure 9.

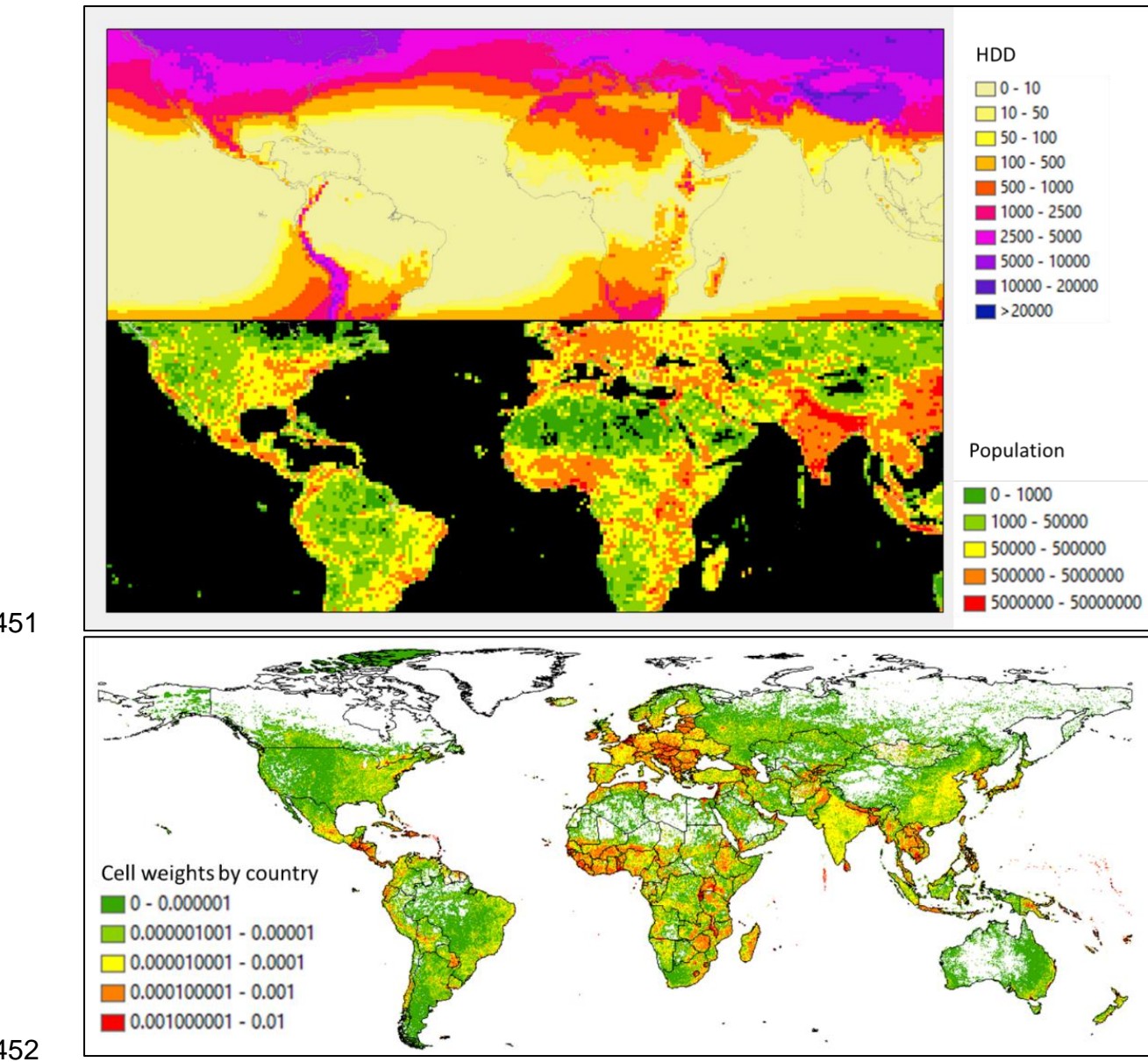



**Figure 9 – Coupling HDDs (top) and population density (middle) as a proxy (bottom) to downscale residential emissions. Data are for the year 2020.**


## 6. Results

The purpose of this work was to describe the methodological improvements included in EDGAR v8.0 linked to the update of the spatial data used to downscale country- and sector-specific emissions. In addition, a specific focus is dedicated to case studies showing the relevance of understanding the trends in GHG emissions at the subnational level in order to support the development of regional climate mitigation and adaptation policies (Kuramochi et al., 2020). The reader can refer to Crippa et al. (2023c) for a description of country- and sector-specific GHG emission trends at the global level. In the following sections, insights on the global distribution of GHG emissions and their subnational features are described.

### 6.1. Global greenhouse gas emissions in EDGAR v8.0

Figure 10 shows global GHG emissions in 2022 as a result of the EDGAR v8.0 gridding process, while Figure 11 reports the same emissions at the country and subnational levels. Complementary figures are also presented in the Supplement. The maps in Figures S5–S8 show the trends in global emissions of GHGs and fossil fuel-derived $CO_2$, $CH_4$ and $N_2O$ from 1970 to 2022.

The main strength and novelty of EDGAR v8.0 is related to the production of a global GHG emission database at different levels of granularity to support local, regional and global climate actions. The high-spatial-resolution global maps are available at $0.1° \times 0.1°$ resolution WGS84 (EPSG:4326), about 10 km spatial resolution at the equator, as both emissions and emission fluxes (.txt and .NetCDF files, https://edgar.jrc.ec.europa.eu/dataset_ghg80) fulfilling the requirements of the global atmospheric modelling community but also bridging bottom-up and top-down (mostly satellite-based) GHG emission estimates (see Figure 10).

EDGAR v8.0 allows full flexibility in the aggregation of emissions at the subnational level, thus supporting the analysis of the spatio-temporal variability of the emissions not only at the grid-cell level but also over wider administrative domains, or areas of interest such as urban centres (Melchiorri, 2022). A second key product from EDGAR v8.0 is represented by GHG emissions at the subnational level using the Global ADMinistrative layer version 4.1 (https://gadm.org/download_country.html) at level 1 and the NUTS 2 level for the EU extended geographical domain, as shown in Figure 11.

Looking at province- or city-scale emissions requires not only associating, for example, point sources to the NUTS 3 level but also relying on an approach different from the downscaling of national totals, which may include the use of statistical information available over smaller territorial units. Therefore, considering the current purposes of EDGAR, the NUTS 2 level represents the right balance between the accuracy of the final emission data and downscaling of national totals. The relevance of including not only country-specific details but also subregional information is essential when doing emission data extraction at the subnational level, thus avoiding border issues. Some inventory compilers (Kuenen et al., 2022) report point source information as just points without distributing them over a grid map with a certain resolution. This approach is accurate, since it provides the exact geographical coordinates of individual facilities; however, it does not reduce data extraction issues, since the allocation of a specific point to a certain grid cell may fall at the border of, for example, two or more regions.

Another challenge that we address with this new gridding approach is related to the harmonisation of national and subnational data. Local and regional inventories are often developed independently, thereby undermining the possibility of combining subnational emission data to retrieve the national values. The challenge of using different and unharmonised databases is overcome by the EDGAR database, as users are able to work consistently at both the national and regional levels, thus offering them the possibility of working across different geographical scales. This is achieved through the downscaling of national emission data to subnational data, making use of high-spatial-resolution proxies, as discussed in this paper. In Sections 6.2 and 6.3 case studies in the European, American and Asian domains are discussed more in detail.

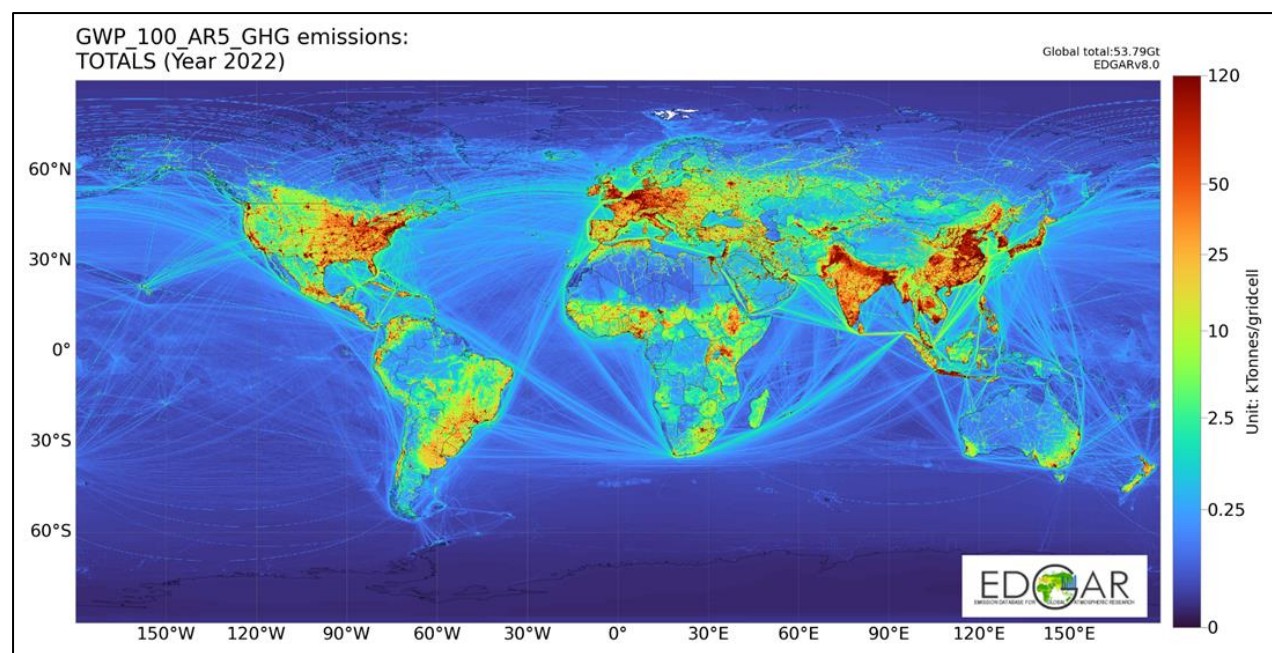

**Figure 10 – Global GHG (expressed in kt CO₂ equivalent) emission map in 2022 from EDGAR v8.0.**

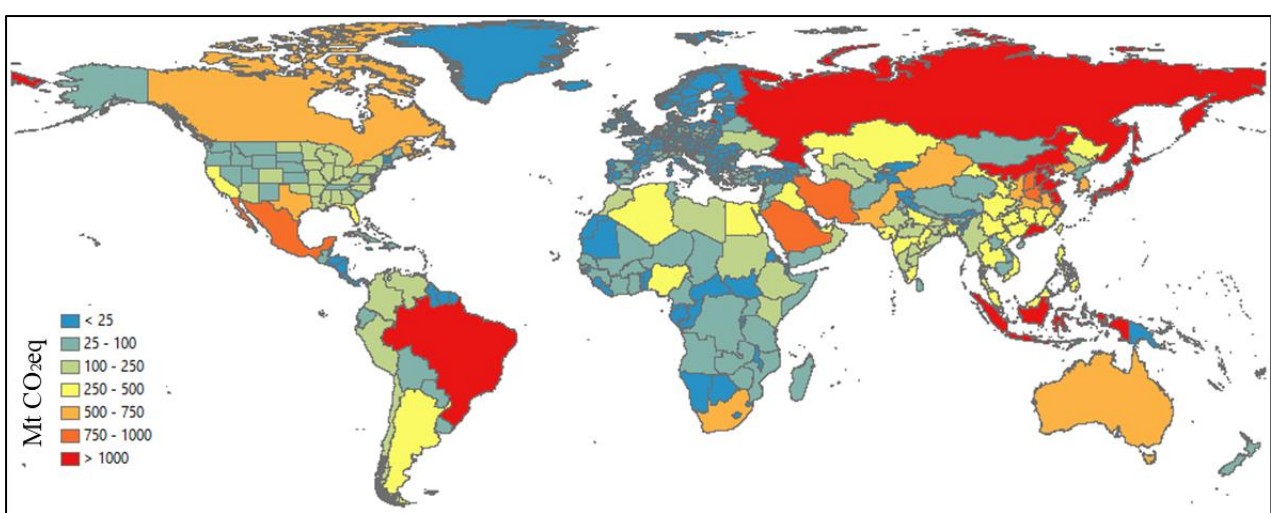

**Figure 11 – Global GHG emissions at the national and subnational levels in 2022 from EDGAR v8.0.**

## 6.2. Subnational emissions: the EU case

Climate and environmental territorial policies require robust and consistent knowledge of GHG and air pollutant emissions at the subnational level (e.g. NUTS 2). No subnational official reporting is available and the high-spatial-resolution data available from EDGAR fill this knowledge gap. EDGAR subnational GHG emissions are used as a reference by the European Commission in cohesion reports (European Commission, 2022), the European semester process and climate action territorial analysis. Figure 12 shows how GHG emissions at the NUTS 2 level changed between 1990 and 2021 in absolute, per capita and per gross domestic product terms. Out of 242 EU regions, 155 regions have shown a downwards trend in emissions since 1990, and 206 and 204 regions have done so since 2005 (on average –1.27 % per year) and 2010 (on average –1.35 % per year), respectively. However, in 2021, only 34 regions

achieved GHG emissions of less than 5 t CO$_2$equivalent/person, which is the average value
needed to achieve the 2030 EU climate targets. The sectors contributing most to total EU GHG
emissions in 2021 are power generation (27 %), industry (23 %), transportation (20 %),
buildings (14 %) and agriculture (11 %), showing that the different regions in the EU have
different transition challenges. For example, when looking at the NUTS 2 level (see Figure 12,
bottom middle panel) the transport sector is often the sector with the largest contribution at the
regional level, in particular in rural regions of Spain, France, Italy and Germany. Figure 12
(bottom right panel) also shows the share of GHG emissions arising from small-scale
combustion (buildings sector) at the NUTS 2 level, highlighting several regions for which this
sector contributes more than 15–20 % to the regional total.

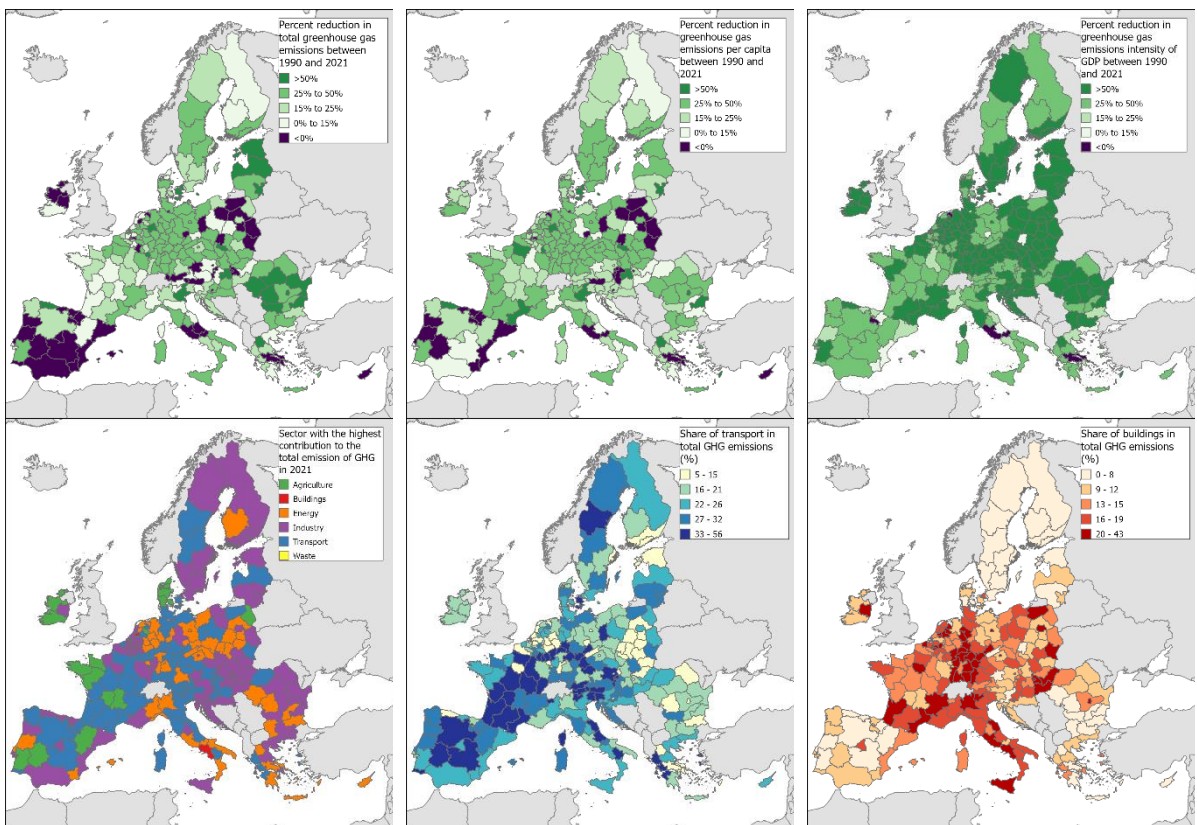


**Figure 12 – Relative change in EU GHG emissions by NUTS 2 level between 1990 and 2021 (top panels).**
**Sectoral contribution to EU GHG emissions by NUTS 2 level in 2021 (bottom panels). The sector with the**
**highest contribution in 2021 for each NUTS 2 region is shown in the map in the left panel. The contribution**
**of GHG emissions from transport (middle panel) and buildings (right panel) to total emissions in 2021 in**
**the EU by NUTS 2 level is also shown.**

## 6.3. Subnational emissions in the United States, China and India

EDGAR v8.0 also includes GHG emission estimates at the subnational level for the United
States (i.e. estimates for each US state, Figure 13) and for each Chinese province and Indian
state (Figure 14). Based on our analysis, Texas emitted 11.5 % of the total US GHG emissions
in 2022, followed by California with a contribution of 7.7 % and Florida with a share of 4.6 %.
In 1990, Texas and California were the most emitting states, followed by Ohio, Pennsylvania

and Illinois. Over the past three decades, the sector with the highest share of GHG emissions at the state level over the United States has changed, with a shift from power generation and industry towards transport (see Figure 13).

In 2022, the five most emitting Chinese provinces contributed around 40 % of China's total GHG emissions. These were Shandong (8.9 % of the country total), Guangdong (8.4 %), Jiangsu (7.4 %), Hebei (6.6 %) and Nei Mongol (6.5 %), findings consistent with other studies addressing provincial $CO_2$ and GHG emissions in China (Jiang et al., 2019; Zhang et al., 2020). In 1990, the top five emitting provinces were Shandong (8.1 %), Hebei (6.5 %), Jiangsu (6.2 %), Henan (5.9 %) and Nei Mongol (5.8 %), contributing around 30 % to China's total GHG emissions.

In 2022, five Indian states contributed around 50 % of the country's total GHG emissions, namely Maharashtra (11.8 %), Tamil Nadu (11.7 %), Uttar Pradesh (8.1 %), Gujarat (8.0 %) and Chhattisgarh (6.6 %). In 1990, the most emitting Indian states were Tamil Nadu (18.4 %), Maharashtra (9.5 %), Uttar Pradesh (9.3 %), West Bengal (6.6 %) and Andhra Pradesh (6.0 %). Compared with the US and European cases, the picture is different over the Asian domain in terms of the top emitting sectors at the subnational level (Figure 14). The effect of India's economic growth and its transition from an agricultural economy to a more industrialised economy can be seen in Figure 14 (right panels). As a result, the sectors with the highest share of GHG emissions changed from agriculture (in 1990) to energy and industry (in 2022) over China and India, with the exception of a few regions (e.g. Tamil Nadu, Assam, Jammu and Kashmir, Uttarakhand) that still had an agriculture-based economy in 2022. This type of information and analysis is instrumental for the definition of effective sector-specific climate change mitigation actions at the subnational level.

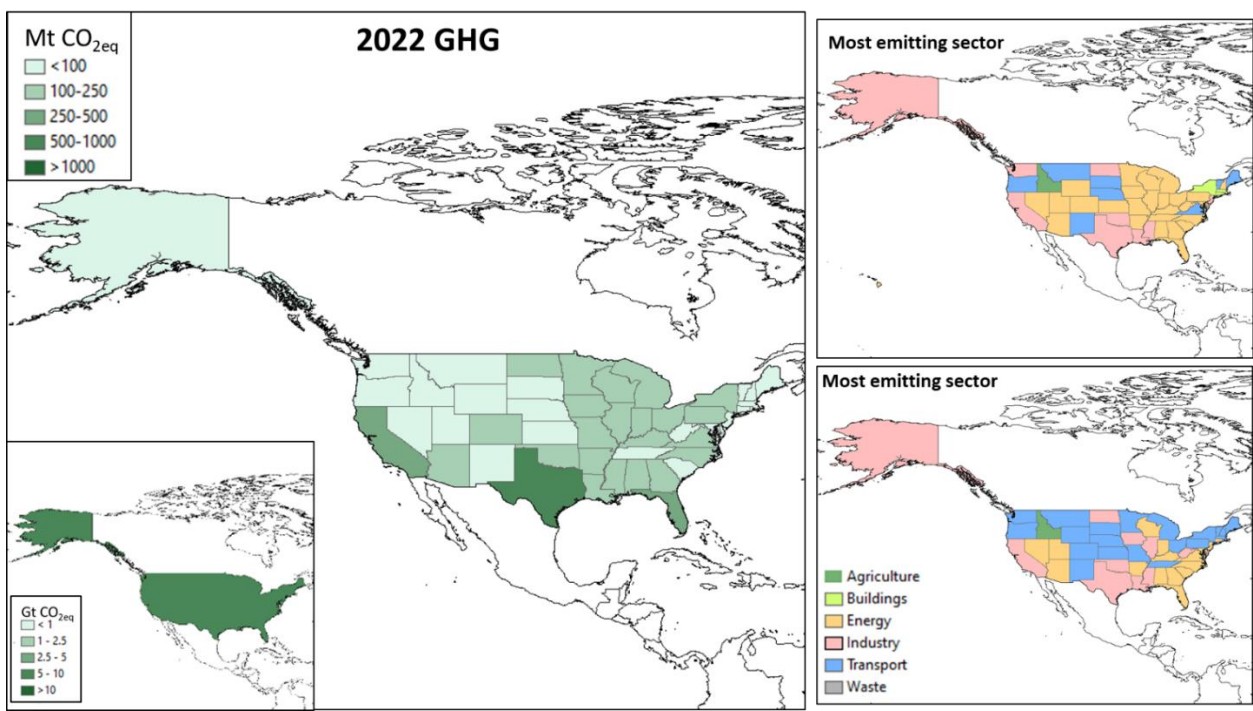

**Figure 13 – 2022 GHG emissions at the subnational level in the United States (left panel) and the sector with the highest contribution to total emissions in 1990 and 2022 for each US state (right panels).**

574

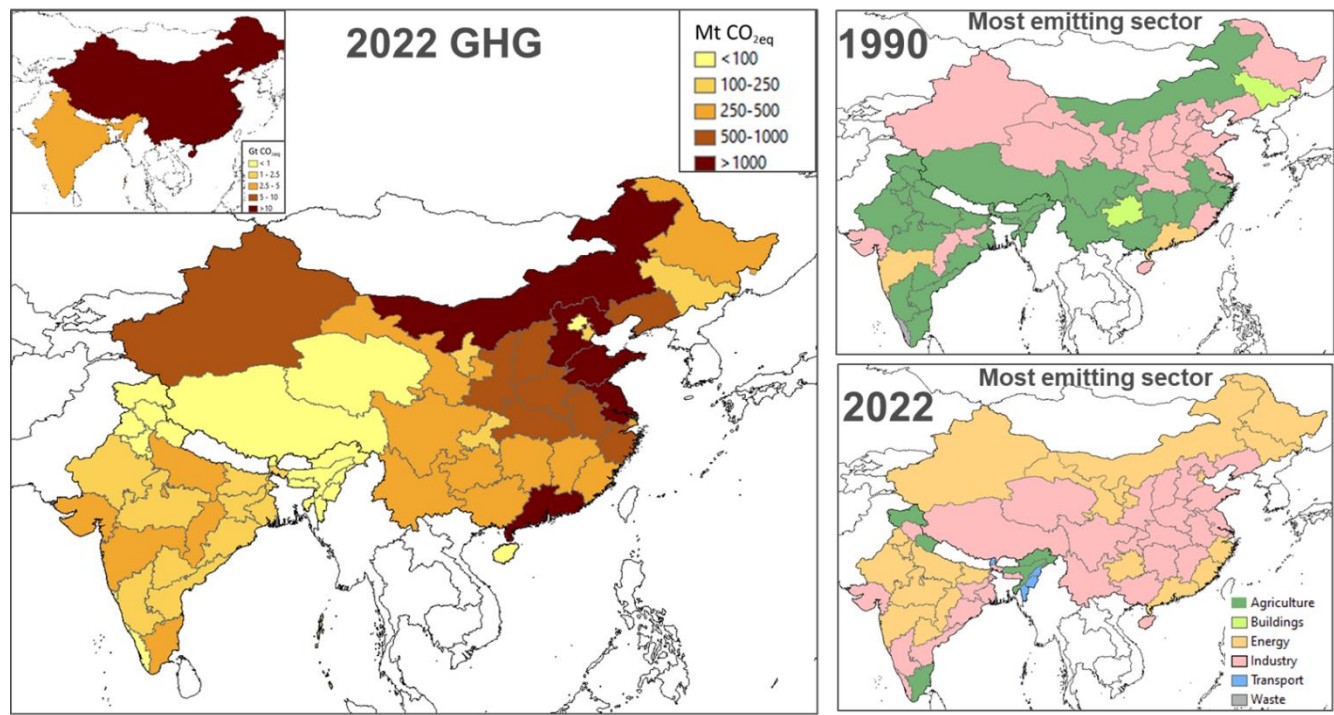

**Figure 14 – 2022 GHG emissions at the subnational level over the Asian domain, with a focus on China and India (left panel) and the sector with the highest contribution in 1990 and 2022 for each Chinese province and Indian state (right panels).**

## 7. Data availability

The EDGAR v8.0 GHG global emission maps can be freely accessed at https://doi.org/10.2905/b54d8149-2864-4fb9-96b9-5fd3a020c224 (Crippa, 2023a). The EDGAR v8.0 subnational emissions can be accessed at https://doi.org/10.2905/D67EEDA8-C03E-4421-95D0-0ADC460B9658 (Crippa et al., 2023b). All data can also be accessed through the EDGAR website at https://edgar.jrc.ec.europa.eu/dataset_ghg80 and https://edgar.jrc.ec.europa.eu/dataset_ghg80_nuts2.

Data are made available as emission grid maps for each species and for total GHGs as .txt and .nc files with emissions expressed in tonnes substance per $0.1° \times 0.1°$ per year. Emission fluxes are available as .nc files and they are expressed in kilograms substance per m$^2$ per second. Emission maps are available as both total and sector-specific emissions.

## 8. Conclusions

Climate targets are often set at the global and national levels; however, their implementation may occur at the subnational level. It is therefore of the utmost relevance to develop subnational GHG emission estimates for policy development and to monitor progress towards climate targets or to evaluate their impacts.

This work summarises the main updates to EDGAR concerning the use of high-resolution and up-to-date spatial information to improve the global geospatial disaggregation of GHG emissions at the subnational level. Having accurate and up-to-date sector-specific global maps of GHG emissions at high spatial resolution ($0.1° \times 0.1°$) is instrumental for the design of effective climate change mitigation options beyond (inter)national climate targets. EDGAR v8.0 spatial proxies include globally consistent spatial data derived, for example, from the

Global Energy Monitor, the GHSL work, satellite-based information for computing HDDs or for identifying hotspots from agricultural activities, STEAM for ship tracking and many other global datasets. The use of satellite data to improve the EDGAR spatial proxies represents a successful cooperation between bottom-up inventory compilers and the Earth observation community and the potential to integrate relevant satellite-based datasets and statistical information. In addition, EDGAR v8.0 integrates spatial information from local databases (e.g. EPRTR for Europe, EIA data for the United States) when including data more detailed than that available in global databases.

Continuous updates and improvements in the spatial data used to downscale national emissions over the global grid are required to accurately represent trends in emission sources and their location. The strength and uniqueness of the EDGAR work arises from its global coverage and consistency in computing and representing emissions for all countries, thus becoming a reference for many countries with limited capabilities to estimate their emissions. However, several challenges are associated with the use of global databases, in particular dealing with the collection of point sources. Therefore, the use of local data, if available, is recommended when performing analysis at the highest spatial resolution (e.g. at the city level).

A further improvement in EDGAR is related to the inclusion of subnational information, representing a unique feature that can address in a consistent way the evaluation of spatial patterns in trends in subnational GHG emissions. Such spatial resolution and subnational sector-specific variability prepares the ground for the production of city-level emission data records, as used, for example, in the Urban Centre Database (https://ghsl.jrc.ec.europa.eu/ghs_stat_ucdb2015mt_r2019a.php). In this paper, a few case studies are presented, with the main focus on the European case where the EDGAR subnational data are regularly used as input to the European semesters and contribute to climate action territorial and cohesion policies through the EU cohesion reports.

The EDGAR v8.0 data release provides an improved GHG dataset that could be useful for air quality modellers but also for policymakers willing to analyse subnational GHG emission patterns. Future EDGAR activities will focus on delivering an updated dataset for air pollutants, including the latest spatial information made available through this work.

## 9. Acknowledgements

We are grateful to William Becker for the thorough review and proofreading of this manuscript. The views expressed in this publication are those of the authors and do not necessarily reflect the views or policies of the European Commission. All emissions, except $CO_2$ emissions from fuel combustion, are from the EDGAR community GHG database comprising IEA-EDGAR $CO_2$, EDGAR $CH_4$, EDGAR $N_2O$ and EDGAR F-gases version 8.0 (2023). The IASI-$NH_3$ catalogue was updated in the framework of the European Space Agency World Emission project (https://www.world-emission.com/). The Université libre de Bruxelles also gratefully acknowledges support from the TAPIR project (Air Liquide Foundation).

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

| Sector and spatial coverage | Old EDGAR proxies | New EDGAR proxies | Details of new EDGAR proxies | Period covered | Data access |
|---|---|---|---|---|---|
| **Power plants (global)** | CARMA v3.0 (no longer available): 2004, 2009, 2014, fuel type derived from plant capacity (assumption) | **Global Coal Plant Tracker / Global Oil and Gas Plant Tracker (Global Energy Monitor)** | Coal, gas | 1970–2050 | https://globalenergymonitor.org/projects/global-coal-plant-tracker/ and https://globalenergymonitor.org/projects/global-gas-plant-tracker/ (2022) |
| | | **Global Power Plant Database v1.3.0** | Biomass, other, oil | | https://datasets.wri.org/dataset/globalpowerplantdatabase |
| | | **US EIA** | USA power plants, all fuels | All | https://atlas.eia.gov/datasets/eia::power-plants/explore?location=41.629235 %2C-118.496000%2C3.79 |
| | | **CARMA v3.0** | Autoproducers, missing countries | 2004, 2009, 2014 | http://carma.org/ |
| **All other industries (Europe)** | EPRTR v4* | **EPRTR, v18** | All industries and waste plants (with the exception of power plants, iron and steel plants, and coal mines) | 2007–2017 | https://www.eea.europa.eu/data-and-maps/data/member-states-reporting-art-7-under-the-european-pollutant-release-and-transfer-register-e-prtr-regulation-23/european-pollutant-release-and-transfer-register-e-prtr-data-base/eprtr_v9_csv.zip |
| **Iron and steel (global)** | In-house EDGAR | **Global steel plant tracker (Global Energy Monitor)** | | 1970–2050 | https://globalenergymonitor.org/projects/global-steel-plant-tracker/ |

| | | | | | |
|---|---|---|---|---|---|
| **Coal mines (global)** | USGS-derived proxies, Global Energy Observatory (China) | **Global Coal Mine Tracker (Global Energy Monitor)** | Brown and hard coal, surface and underground | 1970–2050 | https://globalenergymonitor.org/projects/global-coal-mine-tracker/ |
| | | **Global Energy Monitor + EIA** | United States all fuels, more precise opening and closing years | 1970–2050 | https://atlas.eia.gov/datasets/eia::coal-mines-1/explore |
| | | **EDGAR old proxy** | For missing countries | Key years | |
| **Flaring (global)** | NOAA-NDGC (2015) VIIRS data (https://www.ngdc.noaa.gov/eog/viirs.html) | *Global Gas Flaring Tracker Report* (2023) | Used for both venting and flaring activities | 2012–2022 | https://www.worldbank.org/en/programs/gasflaringreduction/global-flaring-data |
| **Small-scale combustion (global)** | GHSL (1975, 1990, 2000, 2015) | GHSL data package 2023 + HDDs from ERA5 | For all fuels | Population every 5 years from 1975 to 2030; HDDs every year from 1970 to 2022 | https://ghsl.jrc.ec.europa.eu/ghs_pop2023.php and https://cds.climate.copernicus.eu/cdsapp#!/dataset/reanalysis-era5-single-levels?tab=form) |
| **Small-scale combustion in agriculture (global)-rural population** | GHSL (1975, 1990, 2000, 2015) | GHSL data package 2023, including GHS-SMOD R2023A – GHS settlement layers + HDDs from ERA5 | For small-scale combustion in agriculture, which is mostly associated with rural areas | Population every 5 years from 1975 to 2030; HDDs every year from 1970 to 2022 | https://ghsl.jrc.ec.europa.eu/ghs_pop2023.php, https://ghsl.jrc.ec.europa.eu/ghs_smod2023.php, and https://cds.climate.copernicus.eu/cdsapp#!/dataset/reanalysis-era5-single-levels?tab=form) |
| **Intensive livestock and fertiliser-manufacturing industries (global)** | Livestock density maps | European Space Agency world emission project + intensive livestock point sources were taken from EPRTR v18 for Europe | For intensive livestock and fertiliser industry + gap filling with livestock density map | 2008–2022 | https://www.world-emission.com/ |
| **Gap filling of industrial activities (global)** | Population based | Built-up for non-residential areas from GHSL data package 2023 | It is used entirely when no information is available or for attributing a fraction of | Every 5 years from 1975 | https://ghsl.jrc.ec.europa.eu/ghs_buS2023.php |

| International shipping | In-house EDGAR proxy based on long-range identification and tracking and Wang et al. (2007) and Alessandrini et al. (2017) | STEAM | Based on $CO_2$ emissions for multiple vessels and multiple years | 2000–2018 | Jalkanen et al. (2012) Johansson et al. (2017) |
| | | | emissions that is not allocated to point sources | to 2030 | |

805
806