# Peer review of "Insights on the spatial distribution of global, national and sub-national GHG emissions"

_Earth System Science Data, 2023_

## Author Comment (AC1)

Reviewer 2 (https://doi.org/10.5194/essd-2023-514-RC2)

The paper presents a comprehensive overview of the spatial proxies used for EDGAR v8, specifically describing the updates that were made compared to earlier versions. Since EDGAR is a widely used and recognized source of emission information, this contribution is a welcome addition to the scientific community.

The Authors acknowledge the comments made by Reviewer 2 which helped improving the quality of the manuscript. Hereafter, we provide point by point replies in red to each comment.

I would recommend to publish this article but only after a number of comments have been addressed:

- The abstract contains a number of (nearly) copy & pasted sentences from the text, in particular from the introduction. For instance, the first 3-4 sentences of the abstract and the introduction are almost the same. It can probably be summarized in the abstract a bit more.

  The abstract has been revised to avoid duplication of sentences and to best summarise the relevance of this work.

- The introduction is not written very clearly, there seems to be a mix up of introductory sentences and sentences saying what this study is about throughout this section. I believe most necessary elements are there but the structure of this part is lacking, would recommend to rewrite this with a clear structure: introduce the topic and main issues there are, and then in the second part explain what this study is adding to that. See also some of the specific comments below.

  The introduction has been revised in terms of structure and language.

- When the concept of emissions by NUTS region is introduced, there is a lot of attention given to the possible use of this inventory for regional policy analysis. However, most of the information was already available in the EDGAR inventory in its gridded form in previous versions. In fact, the addition of the NUTS regions to the emissions means these emissions by NUTS region can be quantified more precisely. In addition, the use of (often) global proxies makes that the NUTS2/3 region emissions are an approximation. For instance in the EU each country has its own (gridded) emission inventory which likely gives different NUTS2 level emissions from the EDGAR results presented here. The presented approach is however useful to provide a default if such national data is not available (which will be the case in many regions in the world). It would be good to add some reflections to the paper on the constraints of using subnational emission information, perhaps including some of the considerations mentioned here.

  The purpose of our work is to provide readily available emissions at sub-national level estimated in a consistent way for all countries. Although we agree with the Reviewer's comment stating that the EDGAR gridded (and sub-national) emissions may represent an approximation, we also think that even for Europe NUTS2 emissions are not calculated for all EU countries making use of statistics available at NUTS2 level and using NUTS2 specific spatial proxies. No consistently compiled inventory of GHG

emissions exists at NUTS2 level for Europe, but only gridded data representing the downscaling of national emission totals through specific spatial proxies.

We acknowledge the Reviewer's comment and we included the following text in the manuscript:

'The purpose of our work is to provide readily available emissions at sub-national level estimated in a consistent way for all countries. The EDGAR data may represent an approximation for those countries with developed statistical infrastructure (e.g. those including sub-national statistics and very precise spatial proxies), however, they provide a default if such data is not available, as it is the case for many countries in the world.'

- The paper would benefit from a language review. This does not affect understanding of the article, but the use of proper English language should be ensured as it will make the paper better readable. Examples include "multi vessels information" (line 336), "accordingly with" (line 92), but also for instance the use of commas where not needed such as in line 132 (but also in other places in the paper).

The paper has been entirely reviewed by an English mother tongue scientist to improve the correctness of the language.

More specific comments include the following:

- In the abstract some description of the results is missing. E.g. line 29-30 says "the relevance is assessed". But what was the outcome? I think that should be summarized in the abstract.

The abstract has been revised entirely revised following the Reviewer's suggestions.

- Abstract line 29: "main world countries": rather use something like "other large countries"

'Main world countries' was changed to 'other high-emitting countries'.

- Line 41: please include here that EDGAR follows this approach of estimating country level emissions and then gridding them using spatial proxies. This may be trivial for some but not for everyone.

Line 43-44 "although the resolution of underlying spatial information (…) may be higher". I do not see the relevance of this addition at this point in the text?

The sentence has been modified to include the two previous comments of the Reviewer, as following:

'The Emissions Database for Global Atmospheric Research (EDGAR) provides global greenhouse gas (GHG) and air pollutant emissions over the global gridmap at 0.1x0.1 degree resolution, obtained through a downscaling process of national emissions using high-resolution spatial data.'

- Line 46 "to weight national inventories" This is not clear, but I think what you mean is here to disaggregate national emissions to the grid? And "weight" should be "weigh" here.

  The sentence has been changed as following:

  'The development and maintenance of the EDGAR gridmaps is essential since several regional and global databases rely on the EDGAR emission gridmaps to disaggregate national emissions to the grid.'

- Line 48-49 "EU Member States" should rather be "Parties to the LRTAP Convention", as CEIP is not related to EU but to the wider LRTAP Convention.

  Change implemented.

- Lines 50-52 and 63-66 say the same thing twice. Suggest to integrate the part of lines 50-52 with the paragraph in lines 63-66.

  The 2 sentences have been merged as following:

  'This work is an update of previous EDGAR publications dealing with spatial data (Janssens-Maenhout et al., 2019; Crippa et al., 2021), and describes all the new developments for the spatialisation of the emissions from EDGARv8.0 onwards, focusing on high emitting sectors such as power plants and industrial activities, but also on more diffuse sources such as residential activities.'

- Line 54-56: again seems to be almost 1-to-1 copy of this sentence in the abstract

  This sentence has been removed from the abstract.

- Line 58: "power plant" should be "power plants" (plural)

  Change implemented

- Line 59: "distributed" I suppose that here you mean "diffuse"?

  'Distributed' has been replaced with 'diffuse'.

- Line 89: "weight" should be "weigh" (same typo also in line 177)

Change implemented.

- Line 91: Heaviside function: please explain or add reference

The Heaviside function is a unit step function, whose value is zero for negative arguments x < 0 and one for positive arguments x > 0. This is now clarified in the manuscript.

- Line 114: the correct sub-national information: what information? Please be specific.

The sentence has been modified as following:
'A key methodological advancement in the EDGAR gridding system is the inclusion of sub-national attributes for each spatial data and in particular for each point source. This implies attaching to each point not only its exact location expressed in longitude and latitude, but also the related NUTS2 (Nomenclature of territorial units for statistics) code (EUROSTAT, 2021) for Europe or the Global ADMinistrative layer at level 1 (GADM version 4.1).'

- Line 118-119: NUTS3 is more detailed than NUTS2, and can easily be aggregated to NUTS2 in case needed. So how can the use of a more aggregated NUTS2 (compared to NUTS3) enhance the capability to represent sub-national emissions? Please explain or rephrase if something else is meant (probably the latter).

As stated in the methods section, the choice of providing NUTS2 data instead of NUTS3 represents the right balance between accuracy of the final emissions and downscaling of national totals. We report here below the sentences referring to this concept:

The choice of including NUTS2 rather than NUTS3 information aims at enhancing the capability of a global database such as EDGAR to represent sub-national regional emissions in support of the development of regional policies (e.g. EU Cohesion Reports (European Commission, 2022) or the 2040 Climate Impact Assessment), while compromising with the global dimension of the database…..Moving towards province or city scale dimension starting from national emissions is not only subjected to the association of e.g. point sources to NUTS3 level but also relying on a different approach from the downscaling of national totals, which may include the use of statistical information available over smaller territorial units. Therefore, considering the current purposes of EDGAR the NUTS2 level represent the right balance between accuracy of the final emissions and downscaling of national totals.'

- Line 127-128: "more disaggregated statistics". What is meant here? Please explain and/or give example.

Developing a high-resolution inventory (e.g. at NUTS3 level) may require a different approach from the downscale of national emission totals. In fact, the collection of statistics over smaller territorial units is requested. This concept has been clarified in the manuscript as following:

'Moving towards province or city scale dimension starting from national emissions is not only subjected to the association of e.g. point sources to NUTS3 level but also relying on a different approach from the downscaling of national totals, which may include the use of statistical information available over smaller territorial units.'

- Line 162: add "spatially " between "correctly" and "allocate"

Change implemented.

- Line 163-170: It is known that the point source databases mentioned have some gaps and inconsistencies between each other. Please clarify if these databases have been used "as is" or if any corrections or gapfilling has taken place. It would also be good to add specifically what is the metric used for the proxy map: e.g. power plant capacity, reported emissions (CO2 or other?), or something else.

The following sentences have been included in Section 3 to describe some limitations associated with the use of point source databases:

'However, point source databases are characterised by some limitations, in terms of completeness of the point sources, availability of time series of information, misplacement of data points compared to the real country belonging, etc. In EDGAR v8.0, quality checks procedures are applied to validate the correct location of each point source to the corresponding country or sub-national attribute. Moreover, missing information is completed using assumptions on the time life of power plants (i.e. 40 years) to indicatively attribute opening or closing years for each plant.'

- Lines 175-178: if larger gaps exist, has there been any assessments on the consistency between energy consumption in power plants (from statistics) and the installed capacity as a means of cross-check?

No assessment of the consistency between energy consumption available from the IEA energy balances and the installed capacity has been performed in our work, in particular because the purpose of this work is not to build a power plant emission database. However, Guevara et al. (2024) performed this check when using the Global Energy Monitor data for power plants and they found good agreement in national CO2 emissions from power plants as reported by EDGAR (which is based on international statistics) and plant level inventories.

The following statement has been introduced in the text to address this point:

'No consistency check between CO2 emissions estimated through independent methods has been here performed. However, Guevara et al. (2024) have proven the good agreement between national CO2 emissions from power plants as reported by EDGAR (which is based on international statistics) and plant level inventories.'

- Line 198-199: the number of power plants has grown significantly in the database. Is this only a "real" growth in number of plants, or may it also be related to missing plants in earlier years?

Although historic information on power plants reporting is subjected to higher uncertainty compared to nowadays, we believe that the number of power plants increased over the past 5 decades due to industrialisation and higher request of energy at the global level. This is further confirmed by the sharp increase of energy production and use in international statistics. Moreover, the Global Energy Monitor Power Plant database reports information on power plants well before 1970 (which is the base year of EDGAR emissions), thus showing the availability of historic information as well.

'As a general trend, the number of power plants highly increased from 1970 to 2022 (see also Fig.2) due to the industrialisation process happened over the past 5 decades at the global level, although the number of power plants in 1970 is subjected to higher uncertainty compared to nowadays situation.'

- Line 203 "industrialised regions": probably this refers to the regions like Europe and North America? But other regions are also partly industrialised. Please rephrase accordingly.

  The sentence has been changed as following:

  'As shown in Fig.2, despite the increase in the regional number of power plants, shift towards cleaner fuels is found in historically industrialised regions (such as Europe and North America) together with increased energy efficiency, which lead to stable and lower $CO_2$ emissions (e.g. 13% decrease in Europe between 1970 and 2022).'

- Line 219: manufacturing of what? Probably manufacturing industry is meant, but that is an overarching term for the sectors mentioned thereafter. So if that is correct the word "manufacturing" is obsolete here.

  The word 'manufacturing' has been removed.

- Line 220: "solvents" probably refers to the use of solvents, not the production of them. Please rephrase sentence accordingly.

  Solvents has been replaced with 'use of solvents'.

- Line 238-243: What is used as a proxy in the end? Is that iron/steel plant capacity or another metric? In case capacity is used, please add some wording that this is an approximation as some of the installed capacity may actually not be in use.

  We confirm that the installed capacity was used to weigh the proxy of iron and steel plants. The following clarification has been introduced in the manuscript:

  'The installed capacity was used to weigh the relative contribution of each iron and steel plant, although it may represent an approximation for the real capacity in use.'

- Line 253: The section is called "Venting and flaring" but in fact is only about flaring except for one mention of "venting" in line 267. Please clarify the differences and similarities between venting and flaring.

  This concept has been clarified also following the comment of Reviewer 1.

'These spatial data were also used as best approximation to spatially distribute emissions from venting, which is the controlled release of natural gas without being burned, although the two activities may not overlap.'

- Line 259-266: This part is a bit long to make a simple point that few countries make up a large part of the emissions. The point is already made in the first line (259-261) so perhaps the second sentence can be removed (no need to sum up countries beyond the top-4).

  The sentence has been modified as following:

  'Global CO2 emissions related with flaring account for 276 Mt in 2022, of which 76% is emitted by 10 countries, namely Russia (18% of the global total), Iraq (13%), Iran (12%) and Venezuela (7%), followed by Algeria, USA, Mexico, Libya, Nigeria and China.'

- Line 281: "fertilizer industries": does this refer to fertilizer production or fertilizer use?

  It refers to fertiliser production, as now mentioned in the text.

- Please clarify, also because the fertilizer production is already covered in the industrial section.

  As explained at line 304-305, for Europe intensive livestock point sources and fertiliser production industries were taken from EPRTRv18.

- Line 282-283: "satellite-derived NH3 point source database": what do point sources refer to in this case? Probably not individual farms, since typically from satellites you get information at ~10km resolution at the very best.

  NH3 hotspots were retrieved from satellite and were attributed to animal farming and fertiliser production emissions, as explained in detail by Van Damme et al. (2018).

  More specifically, point sources are indeed isolated and localised (or cluster of) emitters that have been identified, categorized and quantified analysing IASI oversampled and supersampled distribution. They mainly consist in agricultural and industrial point sources. Oversampling and supersampling averaging techniques allow to construct NH3 distributions at a resolution (typically 1km) beyond the native resolution of the instrument (in our case, 12km at nadir) exploiting the fact that the location, shape and orientation of the satellite footprints on the ground vary from one orbit to another. We refer here to Van Damme et al., Nature 2018 and Clarisse et al., AMT 2019 for further description on the methodology used to build up the NH3 point source catalogue.

- Line 287: "a fraction": which fraction, how much to the so-called point source?

  A country specific fraction of the emissions is allocated to point sources based on approximate estimates derived from NH3 emissions from IASI. The following explanation is added to the text.

- Line 291: reference/short explanation of previous EDGAR proxy?

A short mention to the previous EDGAR proxy is now reported in the text as following:

'This update represents a significant improvement in representing N related hot-spots (Van Damme et al., 2018) compared to former EDGAR releases which mostly used animal density as proxy (see Table S1), although considering the uncertainty of IASI information of around 50%.'

- Line 321: The ratio between which emissions? This is not clear.

The sentence has been clarifies as following:

'The share of $CO_2$ industrial emissions to the national total over FUAs is typically higher,..'

- Line 337/Section 4: It is mentioned STEAM data were used from EDGAR v6 onwards. Does this imply this section describes the update for shipping in EDGAR v6 which is still the same in EDGAR v8? If yes please make that clear in the text, if not please add changes between different EDGAR versions.

Yes, the STEAM data were already introduced in EDGARv6 and they are still the same also in EDGARv8. We wanted to describe this update related with shipping spatial proxies, since the reference EDGAR papers (e.g. Janssens-Maenhout et al.; 2018, Crippa et al., 2018) describing the spatial proxies of EDGAR were prior to this update which we need to document. The statement has been clarified as following:

'Since EDGARv6.0, international shipping emissions are distributed using the STEAM (Ship Traffic Emission Assessment Model) model from the Finnish Meteorological Institute (Jalkanen et al., 2012; Johansson et al., 2017) and the same spatial distribution is kept also in EDGARv8.0.'

- Related to the previous comment, which version of STEAM emissions is used? And is the same version of those STEAM emissions used in EDGAR v6, (v7), and v8?

The STEAM3 model was used and it is now cited in the text.

For the second part of the question please refer to the previous answer.

- Line 341-344: The information on SECAs, NECAs, etc. and the impact on emissions is already included implicitly in the emissions calculated by STEAM. So why does it needs to be presented here and presented as a future update of EDGAR?

EDGAR shipping emissions are distributed over all routes (thus also covering SECAs and NECAs areas). No weighting factor is applied specifically to SOx or NOx to downscale emissions over these regulated areas, while emissions are weighted using $CO_2$ emissions. This is now clarified in the text as following:

'Information on Emission Control Areas (ECAs), and in particular on sulphur emission control areas (SECAs) and $NO_x$ emission control areas (NECAs), are not yet included since emissions are weighted based on CO2 related emissions, while it represents one of the future updates of EDGAR.'

- Line 351: comparison to EDGAR v5 refers to the last version which was not yet using STEAM emissions, at least that is how it is understood. But what was the proxy used then? Please add that to inform the reader what the comparison shows.

  The full list of spatial proxies used in EDGAR releases prior to EDGARv8 (or EDGARv6 in the case of shipping) are reported in Table S1. However, a clarification for shipping is also now included in the manuscript as following:

  'EDGARv5 used an in-house EDGAR proxy based on Wang et al. (2008) improved with LRIT (Long-Range Identification and Tracking) information (Alessandrini et al., 2017) for European seas, as described in Janssens-Maenhout et al. (2019).'

- Line 358: remove "also"

  Change implemented.

- Line 442: Global administrative layer: reference?

  The reference for the Global ADMinistrative layer is version 4.1 (https://gadm.org/download_country.html), as now mentioned in the manuscript.

- Line 526-527: Implementation may occur at subnational level, but also in many cases at national level.

  Change implemented.

- Line 534: "national" can be "(inter)national"

  Change implemented.

- In the acknowledgements, the authors seem to acknowledge mainly themselves and a range of the EDGAR databases which are basically the dataset corresponding to this paper. This reads very strange, and does not seem appropriate.

  Acknowledgements have been modified removing the first sentence related with the EDGAR team.

References

Alessandrini, A., Guizzardi, D., Janssens-Maenhout, G., Pisoni, E., Trombetti, M., and Vespe, M.: Estimation of shipping emissions using vessel Long Range Identification and Tracking data, Journal of Maps, 13, 946-954, 10.1080/17445647.2017.1411842, 2017.

Guevara, M., Enciso, S., Tena, C., Jorba, O., Dellaert, S., Denier van der Gon, H., and Pérez García-Pando, C.: A global catalogue of CO2 emissions and co-emitted species from power plants, including high-resolution vertical and temporal profiles, Earth Syst. Sci. Data, 16, 337-373, 10.5194/essd-16-337-2024, 2024.

Kuenen, J., Dellaert, S., Visschedijk, A., Jalkanen, J.-P., Super, I., and Denier van der Gon, H.: CAMS-REG-v4: a state-of-the-art high-resolution European emission inventory for air quality modelling, Earth Syst. Sci. Data, 14, 491–515, https://doi.org/10.5194/essd-14-491-2022, 2022.

Van Damme, M., Clarisse, L., Whitburn, S., Hadji-Lazaro, J., Hurtmans, D., Clerbaux, C., and Coheur, P.-F.: Industrial and agricultural ammonia point sources exposed, Nature, 564, 99-103, 10.1038/s41586-018-0747-1, 2018.

Wang, C., Corbett, J., and Firestone, J.: Improving Spatial Representation of Global Ship Emissions Inventories, Environmental science & technology, 42, 193-199, 10.1021/es0700799, 2008.

---

## Author Comment (AC2)

**Reviewer 1 (https://doi.org/10.5194/essd-2023-514-RC1)**

General comments

In this study the development of an updated GHG emissions database, EDGARv8.0, is outlined and supplemented with regional case studies. Updated emissions inventories, point source data, shipping emissions, and proxy methodologies are explored. This effort is aimed to improve the accuracy and consistency of the spatiotemporal distribution of emissions at national and subnational scales. The goal is to better inform climate mitigation and adaptation policy and assist climate modelers in understanding the impact of emissions on the earth and atmosphere.

This paper is rightfully within the scope of ESSD and should be published after minor revisions.

The Authors acknowledge the comments made by Reviewer 1 which helped improving the quality of the manuscript. Hereafter, we provide point by point replies in red to each comment.

Specific comments

Line 69: It is not very clear what is meant by built-up surface information from GHSL. It would be helpful to explain this or refer to some source.

The definition of by built-up surface information and non-residential areas follows the INSPIRE directive: "*A Building is an enclosed construction above and/or underground, used or intended for the shelter of humans, animals or things or for the production of economic goods. A building refers to any structure permanently constructed or erected on its site.*"

This definition of built-up necessarily applied to remote sensing derived products excludes underground structures, while it includes temporary settlements as associated to slums, rapid migratory patterns, or displaced people because of natural disasters or crises. The residential use of built-up areas is defined as dominantly for housing of people, including mixed-use buildings (i.e. having offices or shops occupying part of the floor space). Therefore, "non-residential" indicates industrial or commercial facilities, warehouses, infrastructural nodes etc., which are commonly considered not suitable for residential use.

This concept is now clarified in the text as following:

ii) development of a gap-filling method for non-population based sources using built-up surface information[1]
* * *
[1] This information is compliant with the definition of 'building' as per the 'Infrastructure for Spatial Information in Europe', INSPIRE directive, https://inspire.ec.europa.eu/id/document/tg/bu) for non-residential areas (i.e. industrial or commercial facilities, warehouses, etc.) from the Global Human Settlements Layer (GHSL)

Line 141: How does EDGAR harmonize subnational and national data? Is there some scaling of the subnational data to match the national totals.

A downscaling procedure of national emission totals is applied to obtain gridded (at 0.1x0.1 degree resolution) and sub-national data in EDGAR. Therefore, the sum of sub-national data matches the national values. This concept is clarified as following:

'The challenge of using different and not coherent databases is overtaken by the EDGAR database, being able to consistently work both at the national and regional level, thus offering the user the possibility to work across different geographical scales. This is achieved through the downscaling of national emissions to sub-national data making use of high-spatial resolution proxies, as discussed in this paper.'

Line 148: What is meant by legal site?

'Legal site' has been changed to 'legal address'. The legal address is the place where a company/industry/plant is registered legally but it does not necessarily coincide with the physical location of the company/industry/plant.

Line 253: Can also explain in this section that venting is the release of flare gas (e.g., natural gas) without burning, which is distinct from flaring. Is venting included as an emission source in EDGAR (for CH4)?

The text has been changed as following:

'These spatial data were also used as best approximation to spatially distribute emissions from venting, which is the controlled release of natural gas without being burned, although the two activities may not overlap.'

EDGAR includes CH4 emissions from venting.

Line 327: Is the impact of using these new gap-filling proxies implemented in other databases or validated through other studies?

The use of the non-residential built-up surface information developed by the Global Human Settlements Layer (GHSL) represents a key novelty in the field of global emission inventories. However, similar methodologies are already applied in regional inventories, such as in Europe (Kuenen et al., 2021) where for the area source emissions, the CORINE land-use dataset was used to spatially allocate emissions to areas with industrial activity, thus supporting the validity of this assumption. Therefore, the following statement has been added to the text:

'This methodological assumption is a key novelty of this work due to its application at the global level. However, it is in line with methodologies already applied in regional inventories, such as in Europe (Kuenen et al., 2022) where the CORINE land-use dataset is used to spatially allocate emissions to areas with industrial activity, thus supporting the validity of this assumption.'

Has consideration been given to incorporating emissions at height data as a potential feature? In our recent paper we find that SO2 injection height is a source of inter-model variability, so

having a standardized set of data would be useful for climate models. https://acp.copernicus.org/articles/23/14779/2023/

Emission height is not addressed in this work which is focussing on the spatial characterisation of the emissions. However, we recognise the relevance of this information for atmospheric modelling purposes. Therefore, we added the following sentences in Section 3:

'Atmospheric modellers require information not only the spatial distribution of the emissions but also on the, temporal and vertical distribution of the emissions, as described in Ahsan et al. (2023), Bieser et al. (2011) and De Meij et. al. (2006). For example, De Meij et al. (2006) found that an important role is played by the vertical distribution of SO2 and NOx emissions in understanding the differences between emission inventories on calculated gas and aerosol concentrations. For example, accordingly with the EMEP model, industrial point sources and power plants emissions are injected up to the third level (top up to 184 m), while shipping emissions happen in the first level (top up to 20 m).'

The conclusion section can be improved with some more discussion on potential future works that address the limitations identified in the paper.

Few sentences on the strength and weaknesses of this work have been included.

 Technical corrections

Line 35: Perhaps say "Knowing where emissions are released…"

Change implemented.

Line 113: Can remove "…also represented by…"

Change implemented.

Line 123: Rephrase this to be more coherent, for example "…but also for other countries such as the United States, China, and India, by providing emissions at the state or province level."

Change implemented.

Line 136: The word "cell" here is redundant.

Change implemented.

Line 147: 1970-Present

Change implemented.

Line 161: Can delete "...including the latest available information…"

Change implemented.

Line 271: "2012 to 2022"

Change implemented.

Figure 10: Is this showing CO2 equivalent?

GHG emissions in Figure 10 are expressed in CO2eq. This information has been added to the figure caption.

Line 521: Can use the word species instead of substance.

Change implemented.

Line 543: "…what is available…"

Change implemented.

References

Ahsan, H., Wang, H., Wu, J., Wu, M., Smith, S. J., Bauer, S., Suchyta, H., Olivié, D., Myhre, G., Matsui, H., Bian, H., Lamarque, J.-F., Carslaw, K., Horowitz, L., Regayre, L., Chin, M., Schulz, M., Skeie, R. B., Takemura, T., and Naik, V.: The Emissions Model Intercomparison Project (Emissions-MIP): quantifying model sensitivity to emission characteristics, Atmos. Chem. Phys., 23, 14779–14799, https://doi.org/10.5194/acp-23-14779-2023, 2023.

Bieser, J., Aulinger, A., Matthias, V., Quante, M., Denier van der Gon, H.A.C., 2011. Vertical emission profiles for Europe based on plume rise calculations. Environ. Pollut. 159, issue10, 2935-2946, doi: 10.1016/j.envpol.2011.04.030

De Meij, A., Krol, M., Dentener, F., Vignati, E., Cuvelier, C., and Thunis, P.: The sensitivity of aerosol in Europe to two different emission inventories and temporal distribution of emissions, Atmos. Chem. Phys., 6, 4287–4309, https://doi.org/10.5194/acp-6-4287-2006, 2006.

Kuenen, J., Dellaert, S., Visschedijk, A., Jalkanen, J.-P., Super, I., and Denier van der Gon, H.: CAMS-REG-v4: a state-of-the-art high-resolution European emission inventory for air quality modelling, Earth Syst. Sci. Data, 14, 491–515, https://doi.org/10.5194/essd-14-491-2022, 2022.

---

## Author Response (AR2)

Dear Editor,

In addition to the technical and scientific replies we have provided to the Reviewers' comments on the 25[th] of March 2024, we have followed your suggestion of fully revising the text of the manuscript to enhance its clarity. The paper has been entirely reviewed by the proofreading office of the European Commission and the updated manuscript is now uploaded for your final look.

We are confident that these additional changes have enhanced the quality of this work and will satisfy also your previous concerns.